# Do political incentives promote or inhibit corporate social responsibility? The role of local officials' tenure

Yunyu Wu[ID]*

Department of Business Management, School of Economics and Business Administration, Chongqing University, Chongqing, People's Republic of China

* wuyunyu@cqu.edu.cn

## Abstract

The existing literature on corporate social responsibility (CSR) drivers focuses on firm- and institution-level factors and rarely on the role of political incentives. Public officials control enormous resources in China, and their political incentives substantially shape certain firm behaviors. As CSR is one of the critical measures that the central government uses to evaluate the performance of local government, local officials have the incentive to channel firms into accomplishing their political goals. Correspondingly, local firms may strategically implement CSR to build a good relationship with local governments. This study investigates the impact of local officials' political incentives (measured by tenure) on firms' CSR. Using a panel of publicly listed Chinese firms covering 2009–2019, it documents a U-shaped effect of government officials' tenure on the CSR performance of firms within their jurisdiction. To wit, the firm's CSR decreases first and then increases with the growth of tenure. Moreover, this U-shaped effect will be strengthened in regions with a high priority of gross domestic product (GDP) growth and will be weakened in regions with good market development. In addition, there is no significant evidence that party officials' tenure affects firms' CSR. Overall, this study advances our understanding of the political determinants of CSR in emerging markets.

## Introduction

Corporate social responsibility (CSR) refers to context-specific organizational actions and policies that consider stakeholders' expectations and the triple bottom line of economic, social, and environmental performance [1]. Over recent decades, academic research on CSR has continued to grow. Extensive theoretical research has endeavored to develop a theoretical framework that interprets CSR's underlying drivers from different perspectives, such as institutional and stakeholder theories [2–4]. Meanwhile, empirical studies [5, 6] have strived to find evidence to either support or reject these theoretical arguments. Such studies generally argue that firms' CSR performance is systematically determined by the various individual- [7, 8], firm- [9], and institution-level factors [5, 10, 11]. However, among studies in this inquiry line, very limited attention has been paid to the role of political incentives.

**Data Availability Statement:** The data are available through the databases described in the paper. The data underlying the results presented in the study are available from the RKS website (http://www.rksratings.cn/) and the CSMAR database (http://

cndata1.csmar.com/#/index). Other data on political figures are available from http://cpc. people.com.cn/ and http://baike.baidu.com/. Other economic data are from the publicly available China Statistical Yearbooks and The Five-Year Plans. The above data sources are open to all, and the authors of the present study do not have any special access rights or privileges that other researchers would not have.

**Funding:** The author received no specific funding for this work.

**Competing interests:** The authors have declared that no competing interests exist.

More importantly, the extant research on political incentives and corporate social behavior has been mixed. On the one hand, the literature suggests that political incentives drive firms to engage in more socially oriented organizational practices [12, 13]. On the other hand, it has been argued that political incentives make firms more short-sighted and irresponsible [14]. A possible explanation for these contradictory findings is that the available literature focuses on political promotion or turnover events while ignoring the role of tenure. In the economic literature, tenure is generally considered the length of time an official holds a particular position. Tenure affects officials' behavior and effort for promotion [15]. Evidence shows that officials will exert varying degrees of effort to achieve a particular goal at different stages of tenure, leading to different economic consequences [16] and organizational outcomes [15]. Thus, the term of office may also be a key factor in clarifying the ambiguous relationship between political incentives and corporate social performance.

This paper examined the relationship between local officials' tenure and firms' CSR. Similar to the line of thought in this study is the study by Kong et al. (2021). They examined the impact of political promotion on CSR and found a positive association between local officials' promotion and CSR. However, political promotion events, as a proxy for promotion incentives, may not fully capture the entirety of an official's efforts toward promotion and changes of such efforts. In contrast, tenure better reflects the overall effect of officials' promotion motivation on firms' CSR and how that effect changes.

China offers an ideal setting for investigating the relationship between local officials' tenure and firms' CSR. First, under China's personnel control system, local officials (e.g., provincial governors and party secretaries) serve a limited tenure in a particular position. The formulary tenure of the provincial governor and party secretary is five years for a term, with a maximum of two terms. Generally speaking, officials will be reappointed, promoted, or demoted upon the expiration of the term, depending on their performance during their tenure. Second, China's central government has increased its attention to the issue of the firm's CSR performance [17]. More importantly, CSR is also listed as an essential part of local officials' performance evaluation standards. As a result, the promotion of local officials no longer depends solely on the achievement of economic goals but also on the CSR performance of firms. Third, under China's authoritarian political system, politicians are strongly incentivized because official promotion is the only way for elite politicians to reach the upper echelons of the political hierarchy [18]. Meanwhile, given China's underdeveloped market system and extensive government intervention, local officials have strong incentives to mobilize firms in their jurisdiction to achieve their performance goals and secure promotions.

Specifically, political incentives influence firms' CSR in three ways. First, under China's unique centralized system, local officials have considerable autonomy to implement policies and regulations within their jurisdiction to achieve their political goals regarding CSR [19–21]. Since 2000, the central government has implemented several regulations to incorporate CSR as an essential factor in business operations in China. In 2006, the Shenzhen Stock Exchange (SZSE) and Shanghai Stock Exchange (SSE) issued "The Social Responsibility Guidelines," requiring listed companies to establish social responsibility systems and to form social responsibility reports [17]. Subsequently, in 2009, the central government reformed the performance evaluation standards for local officials and added the social responsibility performance of officials' jurisdiction to the new system. Unlike a single economic performance indicator in the original version, the revised indicator is an aggregate of a group of components, including economic performance, social responsibility, resident welfare, unemployment rate, and relative equity [22]. As a result, the weight of CSR-related indicators remarkably increased while economic indicators experienced a significant drop in evaluation weight [23]. Given the increased weight of CSR in performance evaluation criteria, local officials are motivated to promote

firms' CSR investments through various channels. For instance, they can directly issue commands regarding CSR compliance to state-owned enterprises [24]. For non-state enterprises, such as private firms, local officials can make more frequent inspections and urge them to fulfill CSR (e.g., adopting cleaner production methods) [25]. Also, they can offer more subsidies or tax benefits to companies that perform well in CSR [12].

Second, the achievement of political goals on CSR is also influenced by officials' career concerns. According to career concern theories [26–29], top bureaucrats are largely driven by the outcomes of their mandated tasks. Notably, economic development is still the predominant target in performance evaluation criteria despite the increased weight of CSR [23]. In addition, behavioral research suggests that performing different tasks necessarily entails trade-offs: one goal is accomplished at the expense of another [30, 31]. Previous literature has indicated that the types of goals more likely to attract attention are those tightly coupled with career prospects [32] and easily measurable [33]. In this vein, local officials may be motivated to prioritize local economic growth at the beginning of their tenure and emphasize CSR improvements near the end of their term. At the same time, to enhance the likelihood of promotion, they work to instill their priorities in the companies in their jurisdiction. For instance, early in their tenure, local leaders may relax environmental regulations while meeting the minimum requirements or offer cheaper land and credit to boost short-term gains [34], leading to a decline in CSR. However, towards the end of their tenure, they may use different policy tools, such as enforcing stringent environmental regulations, issuing directives to companies on CSR compliance, providing subsidies or tax benefits for firms' CSR activities, and so on, in order to promote CSR. In short, officials' career concerns cause a shift in the priority of CSR, resulting in a change in the relationship between local officials' tenure and CSR.

Third, resource dependence theory suggests that firm behavior is profoundly influenced by the government because of its control over key external resources [35]. To achieve survival and a competitive position in their industries, firms have a strong incentive to adjust their strategic behaviors to the government's priority. For instance, when the imperative of economic growth is stressed, firms are motivated to pursue short-term profits while ignoring CSR accordingly. Conversely, firms' devotion to CSR significantly increases with the curb on short-term profits when CSR is given priority.

Using a panel of China's A-share-listed companies covering 2009–2019, this study examines the effects of local officials' (i.e., provincial governors and party secretaries) tenure on firms' CSR. Results show a U-shaped relationship between provincial governors' tenure and firms' CSR. The turning point of this curve is about 4.10 years of a governor's tenure. Moreover, this U-shaped curve will steepen in regions with a high priority of GDP growth and flatten in regions with good market development. In addition, I find no significant evidence of a U-shaped relationship between provincial party secretaries' tenure and CSR. This finding suggests that party and government leaders will implement CSR goals differently, given the functional differentiation between the party and government.

This study makes several contributions to the current literature. First, this study adds to our understanding of the state officials' political incentives and determinants of CSR in an emerging economy. This paper shows that as a double-edged sword, political promotion incentives both negatively and positively affect the firms' CSR. This finding is inconsistent with the conclusion of the positive effects of political promotion on CSR derived from the previous research [12]. This distinction suggests that tenure may be vital in investigating the relationship between political motivation and CSR. Moreover, this paper helps understand the multitasking agency problems for government agents. This study shows that driven by career concerns, and government agents are likely to suppress CSR improvement while promoting

local economic growth. This suppression effect is more pronounced, especially when economic growth is given a high priority and the regional market development is poor.

The rest of this paper is arranged as follows: Section 2 gives a review of the local officials' tenure affects firms' CSR and proposes the theoretical analysis and research hypothesis of this paper; Section 3 introduces the method; Section 4 illustrates the method and analyzes the empirical results; Sections 5 and 6 discuss the conclusions, presenting the relevant theoretical and practical implications. This paper ends by elaborating on the limitations and research to be addressed in the future.

## Hypotheses development

### The U-shaped effect of local leaders' tenure on firms' CSR performance

Political incentives based on individuals' career concerns can affect how officials balance multiple political goals and, in turn, influence the firms in their jurisdiction [36]. Specifically, driven by career motives, governmental leaders (i.e., provincial governors) are likely to prioritize economic goals early in their tenure and CSR near the end of their term. Consequently, the CSR performance of companies will decline at first and then rise with the growth of provincial governors' tenure.

Officials have limited attention and resources and may not be able to implement economic goals and CSR performance equally during their limited tenure. Given the high weight of economic goals in the performance assessment scheme, newly appointed government leaders have the impetus to expand investment and respond to promotion assessment by boosting economic growth and controlling CSR, early in their tenure. Specifically, they are likely to favor stimulating economic growth while meeting the minimum requirements of CSR. According to career concern theories, local officials choose their effort levels and distribution of efforts across tasks to maximize their signaled capability to the non-public sectors. Under this framework, to signal their ability to the upper-level supervisors, local officials likely give priority to tasks assigned a high weight by the promotion assessment system [37]. Remarkably, although the degree to which CSR is implemented has become an essential test in the performance evaluation of provincial governors in recent years, economic growth is still the main political incentive for their promotion. Empirical evidence shows that promotions and bonuses of local officials are highly related to regional GDP growth [38]. Compared with other officials at the same level, the higher the overall GDP growth rate during their tenure, the more likely they are to be promoted [39, 40]. Therefore, to increase the likelihood of promotion, provincial governors have strong incentives to promote economic performance and ignore CSR performance due to its relatively loose connection with career advancement [41–43]. Provincial governors' devotion to GDP growth channels firms' attention toward short-term profits and reduce their attention and resource commitment to CSR [15, 44]. As a result, CSR performance is likely to decline in the early stage of the government leader's tenure.

However, the CSR performance will increase as the tenure continues to be extended. According to the law of diminishing marginal utility, the marginal effect of the new economic policy will gradually decrease with the increase of tenure. Correspondingly, the marginal economic performance will decline, and the incentive effect of economic performance on provincial governors will gradually be weakened. In the meantime, rapid economic growth is likely to cause social and environmental problems, such as severe environmental pollution. In order to cope with the promotion assessment, provincial governors urge companies to increase CSR investments to ameliorate severe social and environmental problems. Hence, firms' CSR performance will be improved near the end of the provincial governor's term.

Notably, when local officials serve in a particular position for too long, their room for advancement and likelihood of being promoted will significantly decline. Unlike provincial governors in their first terms, those who continue to serve a second term in their current position are less likely to receive promotions. Chen and Kung (2016) find that 86% of promotions are given to first-term leaders, while the likelihood of promotion significantly decreases after the first term of office [45]. To receive the promotion, first-term governors must strategically focus their attention and resource allocation on tasks closely associated with their career advancement while selectively ignoring other goals [32]. In contrast, continuing governors with fewer chances of promotion are less likely to prioritize promotion-related goals. They likely concentrate on other national goals, such as CSR, to ensure they can get through the remainder of their term well [36]. As such, provincial governors—those who have served in their current positions for more than five years—will continue to emphasize CSR improvements.

Hypothesis 1 (H1). A U-shaped relationship exists between the government leader's tenure and the firms' CSR performance.

Whereas government leaders prioritize distinct goals at different stages of their tenure, party leaders may not. Of note, there is a functional differentiation between government and party leaders, thus leading to the difference in their promotion incentives. Although both provincial party secretaries and governors are in charge of economic and CSR goals, there are variations in their prominence [36]. More often than not, weighting schemes set by the evaluation system assign heavy emphasis to indicators in economic construction for provincial governors. By contrast, provincial party secretaries' work focuses on political tasks, such as social stability, which shares common objectives with CSR [46, 47]. Thus, while government leaders at different stages of their tenure may trade off economic growth and CSR goals, party leaders may not be able to. Given the unambiguous priority of the target for CSR improvements, party leaders are likely to attach great importance to CSR irrespective of their tenure.

Hypothesis 2 (H2). The effect of the government leader's tenure on CSR is more significant than the effect of the party leader's tenure.

## The moderating role of priority of GDP growth

Although government leaders at the same level are evaluated by the same performance assessment criteria, there are variations in economic pressures within them. Under China's unique regional, decentralized authoritarian regime, local governments are required to plan their GDP growth five years ahead in relation to their own situations and set economic growth targets. Remarkably, this target results from negotiations between the central and local governments [48]. Once the targets are decided, local governors try to realize these economic targets to secure their political careers. Consequently, each province's numerical GDP growth target reflects its priority and pressure from the economy. High GDP growth targets indicate local governments' commitment to prioritizing economic growth to some extent. From provincial governors' perspectives, it will be more stressful if the economic targets are high, directly resulting in the neglect of social issues [49]. In contrast, provincial governors with lower economic pressures have more flexibility to achieve their social goals, such as CSR [49].

Under high economic pressures, provincial governors are motivated to boost local GDP quickly [49]. In this context, they are likely to view CSR as a cost or distraction from making profits because CSR is generally regarded as a tool of long-term value creation for businesses and social sustainability [50]. Given the relatively short tenure of government officials and the importance of achieving GDP targets for promotion, governors quickly instill the growth imperative in firms early in their tenure and channel firms' attention toward short-term profits

while reducing their attention and resource commitment to CSR [30]. Thus, the high priority given to GDP growth magnifies the negative effect of the governor's tenure on CSR. At the same time, as a result of the rapid economic growth with the contempt for social and environmental issues, near the end of the term, the governor must urge firms to substantially increase investments in CSR in response to the performance assessment.

Hypothesis 3 (H3). The U-shaped curve between the government leader's tenure and CSR will steepen in regions with a high priority of GDP growth.

### The moderating role of regional market development

The government's influence over firms also varies with the regional marketization level. Although China has made significant progress in marketization, the extent of the progress differs across regions [51]. In some provinces, such as Jiangsu, Guangdong, and Zhejiang, the markets are well-developed, and government intervention is limited [52]. However, in the regions where the market structures are not well developed, the governments still exercise considerable control over firms [53].

In general, the effect of officials' promotion incentives on firm behavior is more profound in regions with a low level of marketization. Less market-oriented regions are usually characterized by a higher degree of government intervention, administrative harassment, and poor legal infrastructures [54]. The inefficient market mechanism expands the power of the government, making the power inequality in the government–business relationship more apparent. Therefore, in less market-oriented regions, firms rely more on the government and are more likely to comply with the political incentives of provincial governors. More specifically, firms will respond more proactively and quickly to the priority of local governors by increasing investments in projects that deliver high and constant profits in the short term or improving CSR performance. It is reasonable to expect that the U-shaped curve between the government leader's tenure and CSR will steepen in less market-oriented regions.

In contrast, the effect of officials' promotion incentives on firm behavior is relatively weak in more market-oriented regions. Generally, more market-oriented regions have a higher quality of market development and better legal infrastructure, such as the protection of property rights and contract enforcement, rendering the distribution of social resources more equitable [54]. In this context, the regional institutional environment creates a lower degree of resource dependence of the firm on the government [55]. Given their less dependence on government resources and smaller power gap in the government–business relationship, firms' decisions largely follow market rules rather than the government's stance. Consequently, firms are less likely to comply with provincial governors' promotion incentives. It is reasonable to claim that the U-shaped curve between the government leader's tenure and CSR will flatten in more market-oriented regions.

Hypothesis 4 (H4). As the regional marketization level increases, the U-shaped curve between the government leader's tenure and CSR will flatten.

## Methodology

### Sample and data sources

I selected Chinese listed firms that disclosed the CSR report on both the SSE and SZSE between 2009 and 2019 and were included in Rankins CSR Rating (RKS) as the original sample. I chose 2009 as the initial study year because many listed firms began disclosure of their CSR information following the implementation of a new policy in that year. I chose 2019 as the termination year of the study because, after this year, the COVID-19 epidemic burst into China, which led to a surge in CSR investment in Chinese companies. Moreover, I exclude financial firms and

firms with negative total assets, negative current liabilities, negative long-term liabilities, or leverage ratios below one. All continuous variables were winsorized in the top and bottom 1% to control for the effect of outliers. After excluding firms with missing information on key variables, the final unbalanced sample contained 985 firms and 6319 firm-year observations.

I first obtained data on CSR from the social responsibility scores provided by RKS (http://www.rksratings.cn/). RKS is a third-party rating agency for CSR in China and one of the major sources for rating the CSR engagement of Chinese firms. Based on the MCTI rating system, RKS adopts a structured expert scoring method, which is triggered by four indicators of macrocosm, content, technique, and industry. The 15 first-level and 63 second-level indicators are constructed to evaluate social responsibility reports comprehensively. Second, I used the China Stock Market and Accounting Research (CSMAR) database (https://www.gtarsc.com/) for firm-level information. The CSMAR database is the primary source for studying Chinese listed firms and provides credible information about companies' backgrounds and financial statistics and has been widely used in management studies. Third, I manually collected the information of provincial party and government leaders from http://cpc.people.com.cn/, a website maintained by People's Daily that provides the most up-to-date news on political figures and events. I double-checked that information against Baidu Baike (http://baike.baidu.com/), a large data source for the curricula vitae of Chinese government officials. In total, 110 governors and 90 provincial committee secretaries served. I collected province-level data from the China Statistical Yearbooks for China's 31 provinces and autonomous regions. Data on the priority of GDP growth was obtained from The Five-Year Plan publicly issued by provinces and autonomous regions. Data on regional market development was gained from the Marketization Index of China's Provinces: NERI Report 2021, jointly compiled by Wang et al.

## Measurements

**Dependent variables.** This study measured a firm's CSR engagement (*CSR score*) using the social responsibility scores provided by RKS, similar to studies that use the KLD (Kinder, Lydenberg, Domini & Co., Inc.) score as an indicator of CSR engagement for US firms [56, 57]. Like KLD, RKS is entirely independent of the companies it rates. The CSR evaluation system established by RKS is based on the Global Reporting Initiative (3.0) framework and incorporates Chinese-specific CSR elements. Its original evaluation data comes from the firms' publicly released social responsibility reports, official websites, and news media. Specifically, RKS sets up 15 first-level and 63 second-level indicators based on the MCTI rating system (See S1 Table) to comprehensively evaluate the degree of a firm's CSR performance. The evaluation system adopts a structured expert scoring method, with a full score of 100. The higher the social responsibility score in RKS, the greater the CSR performance.

Although far from perfect, the RKS data provide a multidimensional assessment of CSR activities conducted by Chinese listed firms based on a variety of sources of information and using consistent and systematic criteria from year to year. The evaluation results have reasonable objectivity, authenticity, and reliability. The RKS data have been widely used in previous CSR studies in China [12, 58–60] as a measure of firms' substantive engagement in social activities with satisfactory results. In addition, validity tests of this measure were also done by studies [49, 61]. These investigations and results on validity tests left us confident that the RKS data are a valid representation of the substantiveness of firms' CSR performance.

**Explanatory variables.** The tenure of a governor (*Tenure of governor*) measures the number of years that a governor has held the same position. Since some leaders assume or leave the office at the beginning or end of the year, I borrow from Bo (1996) that if the governor takes office before June 30, the variable of tenure of governor will be 1 in the current year, or it will

be 1 in the subsequent year [62]; similarly, if the governor leaves office after June 30, the governor will still be considered the leader in the current year. The tenure of the provincial party secretary (*Tenure of party secretary*) was measured similarly.

**Moderating variables.** Regional market development (*MarketDev*) was measured by the Marketization Index of China's Provinces: NERI Report 2021 [53]. The higher the level of marketization, the better the market development, and the lower the government involvement. Regarding local governments' priority for GDP growth, I constructed a dummy variable (*GDP priority*) that equals one if the planned GDP growth stated in a province's five-year plan is higher than the median of planned GDP growth for all 31 provinces in the country, and 0 otherwise, following previous research [63, 64]. This data was collected from the subsection "Main Objectives for Economic and Social Development in the Next Five Years" in the five-year plans for each province.

**Control variables.** I controlled several variables at the firm, province, industry, and individual levels by confirming previous literature regarding CSR research. I first controlled for the conventional firm-specific variables: return on assets, leverage, equity concentration, firm size, firm age, and ownership. Considering that the globalization of firms may increase their pressure on reporting and transparency, I controlled for the foreign income as the percentage of a firm's overseas business income to total operating income [61]. Additionally, the cash flow has been argued to capture better the concept of available resources for discretionary [65]. I, therefore, computed the slack resource as the sum of cash flow from a firm's operating, financing, and investing activities and scaled by total assets to control for firm size [61]. To control for the board-level governance effect, I added four variables: the proportion of independent directors on the board, the board size, the gender of executives, and the age of executives.

Besides, I control for the province's GDP per capita, population growth, and local fiscal revenue to capture the local macroeconomic factors [9]. Given that the overall level of CSR in the industry to which a firm belongs potentially influences, in turn, the substantiveness of its CSR engagement, I controlled for the industry-level CSR, calculated as the mean of the CSR score of the industry [59]. Additionally, the provincial governor's or party secretary's characteristics may be an essential factor. I controlled for the governor and party secretary's age, education, and birthplace. Finally, I added *Industry* dummies to control sector-specific effects. I also included *Year* dummies to control for the omitted variables that vary over time but are constant among the firms. All definitions of the variables used in the analyses are presented in Table 1.

## Estimation model

Given that this study sampled firms from those disclosed CSR reports and excluded firms that did not, there may be a non-random sampling bias stemming from potential systematic differences between firms that disclosed CSR reports and those that did not. Therefore, I use a Heckman two-stage approach to correct for sample selection bias [66]. Heckman two-step procedure requires that at least one variable that does not appear in the second stage should be included in the first stage (i.e., exclusion restriction variable) [67, 68]. This study selected *Policy* as the exclusion variable. *Policy* is a dummy variable that equals 1 if a firm is on the list of mandatory disclosure of CSR reports and 0 otherwise. The reasons for choosing *Policy* as the exclusion variable are as follows.

In response to the central government's call to build a harmonious society and promote sustainable economic and social development, the SSE and SZSE issued the Guidelines on Corporate Social Responsibility of Listed Companies (After this, referred to as the Guidelines) in

**Table 1. Definitions of variables.**

| Variable | Definition |
|---|---|
| **Dependent variable** | |
| CSR score | CSR score is measured by the social responsibility scores provided by RKS. |
| **Explanatory variables** | |
| Tenure of governor | Number of years a governor has been in the post. |
| Tenure of party secretary | Number of years a party secretary has been in the post. |
| **Moderating variables** | |
| MarketDev | Marketization Index of China's Provinces: NERI Report 2021. |
| GDP priority | A dummy variable equals 1 if the planned GDP growth stated in a province's five-year plan is higher than the median of planned GDP growth for all 31 provinces in the country, and 0 otherwise. |
| **Control variables** | |
| Slack resource | The ratio of total cash flow to total assets. |
| ROA | Net profit/average net assets. |
| Leverage | Liabilities/assets. |
| Firm age | The number of years of listing. |
| Firm size | The natural logarithm of total assets. |
| Foreign income | The ratio of overseas business revenue to total business revenue. |
| SOE | 1 = state-owned enterprises, 0 = non-state-owned enterprises |
| Equity concentration | The sum of the shareholding ratios of the top ten shareholders. |
| Board size | The total number of directors on the board. |
| Board independence | The proportion of independent directors on the board. |
| Female executive | A dummy variable equals 1 if there is a female in TMT and board of directors and 0 otherwise. |
| Executive age | Average age of TMT and board members is computed as the sum of the ages of all TMT members divided by the total number of TMT members. |
| Industry-level CSR | The average CSR score of the industry in which the company operates. |
| GDP per capita | Province-level GDP per capita. |
| Population growth | Province-level population growth rate. |
| Fiscal revenue | The natural logarithm of province-level fiscal revenue. |
| Age of governor | Age of governor. |
| Age of party secretary | Age of provincial party secretary. |
| Education of governor | 1 = below bachelor, 2 = bachelor, 3 = master, 4 = PhD. |
| Education of party secretary | 1 = below bachelor, 2 = bachelor, 3 = master, 4 = PhD. |
| Nativeplace of governor | A dummy variable equals 1 if a governor serves in the province of his naturalization or his hometown province and 0 otherwise. |
| Nativeplace of party secretary | A dummy variable equals 1 if a party secretary serves in the province of his naturalization or his hometown province and 0 otherwise. |

succession in 2008. According to the Guidelines, four types of firms are required to disclose CSR reports compulsorily: the SSE's three types of firms (i.e., firms in the 'corporate governance group,' firms listed in foreign stock exchanges, and firms in financial industries), and firms in the 'Shenzhen Stock Exchange 100 Index' in the SZSE [61, 69]. Therefore, it is reasonable to expect that firms on this mandatory CSR disclosure list are more likely to release CSR reports than those not on this list. Remarkably, although the Guidelines require firms on this list to issue CSR reports, it does not require them to spend on CSR. Therefore, *Policy* is directly related to the release of CSR reports yet is not directly related to the level of firms' engagement

in CSR. The first-stage estimation model is as follows:

$$Report_{i,t+1} = \beta_0 + \beta_1 Exclusion_{i,t} + ControLfirm_{i,t} + ControLindustry_{i,t} + \varepsilon_{i,t} \qquad (1)$$

The subscripts i and t denote the firm and year, respectively. The dependent variable *Report* is a dummy variable equal to 1 if a firm discloses a CSR report and 0 otherwise. *Exclusion* represents the 'exclusion restriction' variable (i.e., *Policy*). *Control_firm* is a set of firm-level control variables, including slack resource, return on assets, leverage, firm age, firm size, foreign income, equity concentration, board size, board independence, and executive gender and age. *Control_industry* is the industry-level CSR score. $\varepsilon$ is an error term clustered at the firm level. Furthermore, the year dummies are also included in the model. In the first stage, a Probit model is conducted to predict the likelihood that firms engage in CSR activities comprising the entire sample of firms engaging in CSR activities and those not engaging in CSR activities. The first-stage regression results generated the inverse Mills ratio (*IMR*), which was incorporated into the second-stage regressions. The second-stage estimation models are as follows:

$$CSR\ score_{i,t+1} = \boldsymbol{\beta_0} + \boldsymbol{\beta_1} \boldsymbol{Tenure}_{i,t} + \boldsymbol{\beta_2} \boldsymbol{Tenure}^2_{i,t} + \boldsymbol{\beta_3} \boldsymbol{IMR}_{i,t} + \boldsymbol{Controls}_{i,t} + \boldsymbol{\varepsilon}_{i,t} \qquad (2)$$

$$
\begin{aligned}
CSR\ score_{i,t+1} \\
&= \beta_0 + \beta_1 Tenure_{i,t} + \beta_2 Tenure^2_{i,t} + \beta_3 Tenure_{i,t} \times GDP\ priority_{i,t} \\
&+ \beta_4 Tenure^2_{i,t} \times GDP\ priority_{i,t} \\
&+ \beta_5 Tenure_{i,t} \times MarketDev_{i,t} + \beta_6 Tenure^2_{i,t} \times MarketDev_{i,t} \\
&+ \beta_7 IMR_{i,t} + Controls_{i,t} + \varepsilon_{i,t}
\end{aligned}
\qquad (3)
$$

Eq (2) is used to test H1 and H2, and Eq (3) is used to test H3 and H4. Where *CSR score* is the dependent variable, *Tenure* is the independent variable representing the *Tenure of governor* or *Tenure of party secretary*, and *IMR* represents the inverse Mills ratio generated by the first-stage regression results. *Controls* is a set of the firm-, province-, industry-, and individual-level control variables, and $\varepsilon$ is an error term clustered at the firm level to ensure the robustness of the conclusions. The industry and year dummies are included to control for fixed industry-specific, time-variant characteristics. Considering the potential endogeneities of reverse causality, I lag one year between the dependent variable (t + 1) and the independent variables (t) for all estimations [70].

## Results

### Main results

Table 2 displays the descriptive statistics of all variables for the first-stage and second-stage models, and correlation metrics are represented in S2 Table. Table 3 represents the results for the first-stage Probit model. As predicted, firms on the list of mandatory disclosure are more likely to issue CSR reports than others. The IMR was calculated based on Model 2 in Table 3 and incorporated into the following second-stage regressions.

In Heckman second-stage models, I use a three-step procedure to test H1 and H2 [71, 72]. First, I tested whether the coefficients of the squared terms of the government and party leader's tenure in the ordinary least squares multiple regression analysis were significant and of the expected sign. Second, I carried out Sasabuchi tests [72] to check whether each predicted U-shaped curve had sufficiently steep slopes at each end of the data range. Third, I checked whether the estimated turning points were located within the data ranges. Moreover, I tested H3 and H4 in the following way: To the regressions used to test Hypotheses 1, I added

**Table 2. Descriptive statistics.**

| Panel A: Heckman first-stage variables | | | | | | |
|---|---|---|---|---|---|---|
| **Variable** | **Mean** | **SD** | **Median** | **Minimum** | **Maximum** | **N** |
| Report | 0.241 | 0.428 | 0 | 0 | 1 | 26220 |
| Policy | 0.138 | 0.345 | 0 | 0 | 1 | 26220 |
| Slack resource | 0.041 | 0.075 | 0.041 | -0.199 | 0.246 | 26220 |
| ROA | 0.072 | 0.124 | 0.076 | -0.526 | 0.405 | 26220 |
| Leverage | 0.434 | 0.218 | 0.425 | 0.048 | 0.969 | 26220 |
| Firm age | 10.305 | 6.845 | 9 | 1 | 24 | 26220 |
| Firm size | 21.962 | 1.284 | 21.8 | 19.291 | 25.8 | 26220 |
| Foreign income | 0.104 | 0.199 | 0 | 0 | 0.886 | 26220 |
| Equity concentration | 58.409 | 15.618 | 59.59 | 21.89 | 90.25 | 26220 |
| Board size | 10.04 | 2.531 | 9 | 5 | 18 | 26220 |
| Board independence | 0.379 | 0.071 | 0.364 | 0.25 | 0.6 | 26220 |
| Female executive | 0.908 | 0.289 | 1 | 0 | 1 | 26220 |
| Executive age | 48.444 | 3.2 | 48.5 | 41 | 55.857 | 26220 |
| Industry-level CSR | 33.395 | 13.438 | 37.422 | 0 | 46.535 | 26220 |
| Panel B: Heckman second-stage variables | | | | | | |
| CSR score | 38.991 | 12.089 | 36.5 | 18.86 | 74.95 | 6319 |
| Tenure of governor | 3.019 | 2.057 | 2 | 1 | 9 | 6319 |
| Tenure of party secretary | 3.058 | 2.111 | 2 | 1 | 10 | 6319 |
| GDP priority | 0.449 | 0.497 | 0 | 0 | 1 | 6319 |
| MarketDev | 7.948 | 1.737 | 8.33 | -0.23 | 10 | 6319 |
| Slack resource | 0.053 | 0.07 | 0.051 | -0.199 | 0.246 | 6319 |
| ROA | 0.091 | 0.111 | 0.089 | -0.526 | 0.405 | 6319 |
| Leverage | 0.489 | 0.199 | 0.502 | 0.048 | 0.969 | 6319 |
| Firm age | 12.568 | 6.265 | 13 | 1 | 24 | 6319 |
| Firm size | 22.984 | 1.371 | 22.87 | 19.291 | 25.8 | 6319 |
| Foreign income | 0.095 | 0.182 | 0 | 0 | 0.886 | 6319 |
| SOE | 0.618 | 0.486 | 1 | 0 | 1 | 6319 |
| Equity concentration | 59.503 | 16.158 | 59.83 | 21.89 | 90.25 | 6319 |
| Board size | 10.63 | 2.696 | 10 | 5 | 18 | 6319 |
| Board independence | 0.378 | 0.072 | 0.364 | 0.25 | 0.6 | 6319 |
| Female executive | 0.884 | 0.32 | 1 | 0 | 1 | 6319 |
| Executive age | 49.509 | 3.136 | 49.583 | 41 | 55.857 | 6319 |
| GDP per capita | 10.992 | 0.505 | 11.058 | 9.641 | 11.851 | 6319 |
| Population growth | 5.029 | 2.269 | 4.95 | -0.39 | 10.84 | 6319 |
| Fiscal revenue | 17.237 | 0.754 | 17.296 | 14.798 | 18.612 | 6319 |
| Industry-level CSR | 35.304 | 11.599 | 37.422 | 0 | 46.535 | 6319 |
| Age of governor | 58.246 | 4.063 | 59 | 48 | 65 | 6319 |
| Age of party secretary | 61.191 | 4.255 | 62 | 50 | 70 | 6319 |
| Education of governor | 3.022 | 0.639 | 3 | 2 | 4 | 6319 |
| Education of party secretary | 2.857 | 0.67 | 3 | 1 | 4 | 6319 |
| Birthplace of governor | 0.215 | 0.411 | 0 | 0 | 1 | 6319 |
| Birthplace of party secretary | 0.051 | 0.22 | 0 | 0 | 1 | 6319 |

Note. Panel A exhibits the descriptive statistics of Heckman first-stage variables, including firms that disclosed CSR reports and those that did not. In panel A, *Report* equals 1 if a firm discloses a CSR report and 0 otherwise, *Policy* equals 1 if a firm is on the list of mandatory disclosure of CSR reports and 0 otherwise, and the definitions of other variables are presented in Table 1. Panel B reports the descriptive statistics of Heckman second-stage variables, including only firms that disclosed CSR reports. In panel B, the definitions of all variables are presented in Table 1.

**Table 3. Probit estimates for Heckman first-stage model.**

| Model No. | Model 1 | Model 2 |
|---|---|---|
| Policy | | 4.251*** |
| | | (12.773) |
| Slack resource | 0.383 | 0.497 |
| | (0.894) | (1.210) |
| ROA | -0.110 | -0.020 |
| | (-0.406) | (-0.072) |
| Leverage | -2.250*** | -1.559*** |
| | (-6.126) | (-5.044) |
| Firm age | 0.111*** | 0.073*** |
| | (7.164) | (6.221) |
| Firm size | 1.497*** | 1.009*** |
| | (13.309) | (11.795) |
| Foreign income | -0.696 | -0.471 |
| | (-1.593) | (-1.321) |
| Equity concentration | -0.010** | 0.004 |
| | (-2.084) | (0.992) |
| Board size | -0.013 | 0.002 |
| | (-0.955) | (0.138) |
| Board independence | -0.743* | -0.871** |
| | (-1.745) | (-2.055) |
| Female executive | -0.182 | -0.124 |
| | (-1.436) | (-0.984) |
| Executive age | -0.010 | 0.002 |
| | (-0.475) | (0.101) |
| Industry-level CSR | 0.012*** | 0.004 |
| | (3.684) | (1.406) |
| Constant | -34.072*** | -25.717*** |
| | (-13.565) | (-13.327) |
| Year FE | yes | yes |
| Industry FE | no | no |
| Observations | 26220 | 26220 |
| Wald $\chi$ | 421.5*** | 528.52*** |

Note. Correlation metrics of all variables in the first stage are represented in S2 Table. Robust z-statistics (in parenthesis) are based on the standard errors clustered by firms.

*** $p<0.01$,

** $p<0.05$,

* $p<0.1$

interactions of GDP priority and market development with the linear and squared terms of the governor's tenure. The estimates of the interactive effects provided direct tests for the significance of the difference in the effects of the government leaders' tenure on CSR under different levels of moderators.

Table 4 presents the results of Heckman second-stage models, including the test of H1-4. The results corroborate H1, postulating a U-shaped relationship between the tenure of the government leader and CSR. The coefficients corresponding to the quadratic terms of the government leader's tenure are significant in Models 2 and 4. Sasabuchi tests show that the slope at

**Table 4. Effects of government leaders' tenure on CSR and moderating effects of GDP priority and regional market development (Heckman second-stage model).**

| Model No. | Model 1 | Model 2 | Model 3 | Model 4 | Model 5 | Model 6 | Model 7 |
|---|---|---|---|---|---|---|---|
| Tenure of governor | | -0.599** | | -0.728*** | 0.053 | -0.811*** | -0.237 |
| | | (-2.430) | | (-3.037) | (0.164) | (-3.334) | (-0.712) |
| Tenure of governor$^2$ (H1) | | 0.074** | | 0.089*** | 0.009 | 0.101*** | 0.046 |
| | | (2.275) | | (2.828) | (0.256) | (3.159) | (1.234) |
| Tenure of party secretary | | | 0.122 | 0.061 | 0.290 | 0.022 | 0.241 |
| | | | (0.619) | (0.320) | (1.469) | (0.110) | (1.200) |
| Tenure of party secretary$^2$ (H2) | | | 0.008 | 0.019 | -0.011 | 0.022 | -0.009 |
| | | | (0.339) | (0.856) | (-0.491) | (0.946) | (-0.379) |
| GDP priority | | | | | -1.992*** | | -2.338*** |
| | | | | | (-3.533) | | (-3.943) |
| Tenure of governor*GDP priority | | | | | -1.296*** | | -0.780 |
| | | | | | (-3.026) | | (-1.623) |
| Tenure of governor$^2$*GDP priority (H3) | | | | | 0.152*** | | 0.092* |
| | | | | | (3.231) | | (1.790) |
| MarketDev | | | | | | -0.749* | -0.864** |
| | | | | | | (-1.855) | (-2.087) |
| Tenure of governor*MarketDev | | | | | | 0.239* | 0.261* |
| | | | | | | (1.950) | (1.827) |
| Tenure of governor$^2$*MarketDev (H4) | | | | | | -0.031* | -0.037* |
| | | | | | | (-1.816) | (-1.870) |
| Slack resource | 9.199*** | 9.237*** | 9.156*** | 9.208*** | 9.139*** | 9.314*** | 9.194*** |
| | (3.127) | (3.141) | (3.114) | (3.134) | (3.119) | (3.183) | (3.158) |
| ROA | -0.199 | -0.104 | -0.217 | -0.115 | -0.255 | 0.037 | -0.072 |
| | (-0.106) | (-0.056) | (-0.115) | (-0.061) | (-0.137) | (0.020) | (-0.038) |
| Leverage | -6.015*** | -5.879*** | -6.027*** | -5.879*** | -5.739*** | -5.789*** | -5.619*** |
| | (-3.348) | (-3.275) | (-3.355) | (-3.276) | (-3.212) | (-3.234) | (-3.153) |
| Firm age | -0.047 | -0.051 | -0.047 | -0.051 | -0.056 | -0.048 | -0.054 |
| | (-0.771) | (-0.837) | (-0.770) | (-0.838) | (-0.928) | (-0.793) | (-0.890) |
| Firm size | 4.157*** | 4.150*** | 4.149*** | 4.139*** | 4.101*** | 4.107*** | 4.059*** |
| | (10.428) | (10.432) | (10.414) | (10.418) | (10.317) | (10.318) | (10.194) |
| Foreign income | 0.826 | 0.827 | 0.787 | 0.775 | 0.807 | 0.861 | 0.902 |
| | (0.560) | (0.560) | (0.535) | (0.526) | (0.552) | (0.586) | (0.618) |
| SOE | 1.372* | 1.403* | 1.353* | 1.378* | 1.389* | 1.270* | 1.264* |
| | (1.911) | (1.960) | (1.888) | (1.928) | (1.945) | (1.774) | (1.770) |
| Equity concentration | 0.069*** | 0.069*** | 0.069*** | 0.070*** | 0.068*** | 0.070*** | 0.069*** |
| | (3.139) | (3.160) | (3.152) | (3.177) | (3.128) | (3.212) | (3.160) |
| Board size | 0.123 | 0.121 | 0.122 | 0.121 | 0.113 | 0.117 | 0.109 |
| | (1.370) | (1.350) | (1.367) | (1.348) | (1.271) | (1.312) | (1.229) |
| Board independence | -1.163 | -1.154 | -1.221 | -1.217 | -1.049 | -1.378 | -1.223 |
| | (-0.417) | (-0.414) | (-0.439) | (-0.437) | (-0.380) | (-0.497) | (-0.444) |
| Female executive | 1.275* | 1.276* | 1.245 | 1.236 | 1.255* | 1.233 | 1.256* |
| | (1.667) | (1.674) | (1.629) | (1.620) | (1.653) | (1.618) | (1.658) |
| Executive age | 0.252** | 0.252** | 0.252** | 0.253** | 0.252** | 0.260*** | 0.260*** |
| | (2.524) | (2.525) | (2.532) | (2.535) | (2.532) | (2.610) | (2.621) |
| GDP per capita | 0.414 | 0.345 | 0.271 | 0.162 | -0.284 | 1.552 | 1.228 |
| | (0.399) | (0.336) | (0.261) | (0.159) | (-0.278) | (1.174) | (0.937) |
| Population growth | 0.176 | 0.193 | 0.122 | 0.129 | 0.093 | 0.134 | 0.098 |

(*Continued*)

**Table 4.** (Continued)

| Model No. | Model 1 | Model 2 | Model 3 | Model 4 | Model 5 | Model 6 | Model 7 |
|---|---|---|---|---|---|---|---|
| | (1.316) | (1.424) | (0.891) | (0.933) | (0.671) | (0.983) | (0.719) |
| Fiscal revenue | 1.217* | 1.195* | 1.195* | 1.168* | 0.850 | 2.037*** | 1.829** |
| | (1.959) | (1.922) | (1.923) | (1.879) | (1.356) | (2.692) | (2.417) |
| Industry-level CSR | 0.029* | 0.029* | 0.030* | 0.030* | 0.031* | 0.032** | 0.034** |
| | (1.813) | (1.827) | (1.845) | (1.861) | (1.950) | (1.997) | (2.089) |
| Age of governor | 0.130*** | 0.128*** | 0.101** | 0.094* | 0.064 | 0.082 | 0.052 |
| | (2.797) | (2.589) | (2.060) | (1.830) | (1.213) | (1.596) | (0.986) |
| Age of party secretary | -0.214*** | -0.214*** | -0.248*** | -0.258*** | -0.266*** | -0.259*** | -0.266*** |
| | (-3.299) | (-3.303) | (-3.685) | (-3.793) | (-3.893) | (-3.801) | (-3.897) |
| Education of governor | 0.898** | 0.732* | 0.950** | 0.769** | 0.623* | 0.694* | 0.555 |
| | (2.410) | (1.931) | (2.540) | (2.023) | (1.679) | (1.794) | (1.463) |
| Education of party secretary | -0.273 | -0.218 | -0.328 | -0.291 | -0.082 | -0.414 | -0.159 |
| | (-1.093) | (-0.893) | (-1.273) | (-1.151) | (-0.328) | (-1.580) | (-0.628) |
| Birthplace of governor | -0.454 | -0.601 | -0.261 | -0.365 | -0.836 | -0.450 | -1.033 |
| | (-0.839) | (-1.037) | (-0.470) | (-0.611) | (-1.345) | (-0.743) | (-1.639) |
| Birthplace of party secretary | -2.004 | -1.835 | -2.444* | -2.350* | -1.577 | -2.736** | -1.940 |
| | (-1.565) | (-1.437) | (-1.856) | (-1.779) | (-1.211) | (-2.034) | (-1.471) |
| IMR | 0.633*** | 0.627*** | 0.633*** | 0.628*** | 0.642*** | 0.656*** | 0.673*** |
| | (3.007) | (2.983) | (3.013) | (2.991) | (3.071) | (3.132) | (3.222) |
| Constant | -97.540*** | -95.057*** | -91.864*** | -87.525*** | -74.229*** | -116.984*** | -105.861*** |
| | (-6.844) | (-6.598) | (-6.233) | (-5.934) | (-4.965) | (-5.559) | (-5.081) |
| Industry FE | Yes | yes | yes | Yes | yes | yes | yes |
| Year FE | Yes | yes | yes | Yes | yes | yes | yes |
| Observations | 6319 | 6319 | 6319 | 6319 | 6319 | 6319 | 6319 |
| R-squared | 0.361 | 0.362 | 0.362 | 0.363 | 0.366 | 0.364 | 0.368 |
| F | 13.566 | 12.565 | 12.768 | 11.894 | 10.995 | 10.937 | 10.161 |

Note. Correlation metrics of all variables in the second stage are represented in S2 Table. Robust t-statistics (in parentheses) are based on the standard errors clustered by firms to address potential serial correlations in the residuals.

*** p<0.01,

** p<0.05,

* p<0.1.

the low end of the governor's tenure is negative and significant at the 1% level, and the slope at the high end is positive and significant at the 1% level. The turning point of the estimated U-shaped effect is 4.10 years of the governor's tenure. I computed the 95% confidence interval for this turning point and confirmed that this point lies within the data range. The results of statistical tests of the U-shaped relationship between the government leader's tenure and CSR are reported in S3 Table.

The analyses also support H2, predicting that the effect of the government leader's tenure on CSR is more significant than the effect of the party leader's tenure. The coefficients corresponding to the quadratic terms of the party leader's tenure are insignificant in Models 3 and 4. This finding indicates that the functional differentiation between the party and the government leaders leads to variations in their incentives for the firm's CSR behavior.

The results confirm H3, proposing that the U-shaped curve between the governor's tenure and CSR will steepen in regions with a high priority of GDP growth. Model 7 in Table 4 shows

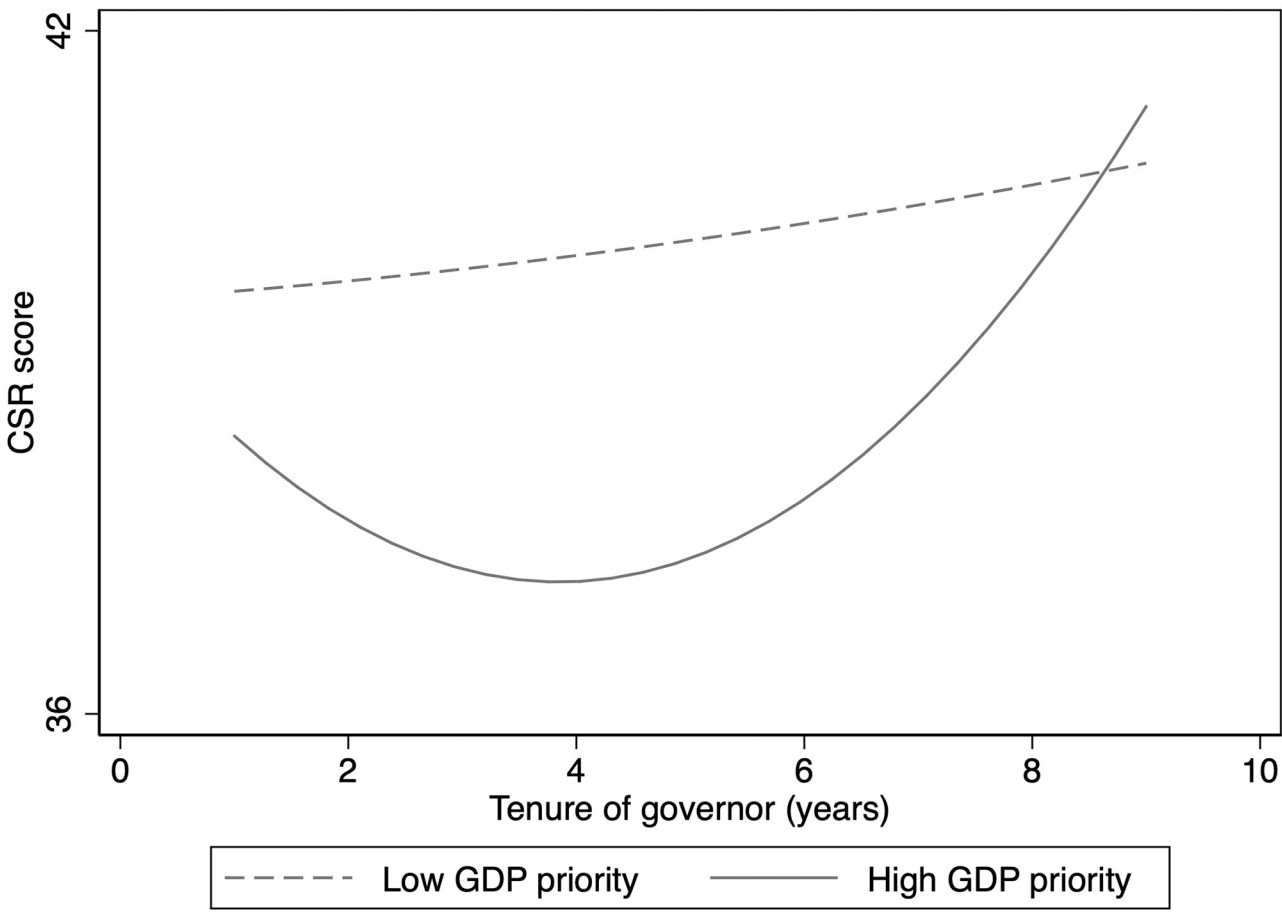

**Fig 1. Moderating effect of regional GDP priority.**

that the interaction of the GDP priority dummy with the squared term of the governor's tenure is positive and significant at the 10% level. This indicates that the high priority of GDP growth strengthens the U-shaped effect of the government leader's tenure on CSR. Furthermore, the turning point of this U-shaped curve is calculated and found to be 3.68 in the case of a high GDP priority. This result indicates that the negative effect of the governor's tenure on CSR will be exacerbated in regions with a high priority of GDP growth. As shown in Fig 1, both the downward and upward sides of the relationship steepen under conditions of high GDP priority. At the same time, Fig 1 also shows that high economic pressures foster the short-sighted behavior of provincial governors. Specifically, high GDP growth targets spur newly appointed governors to boost GDP quickly using various policy tools (e.g., relaxed environmental regulations), thus leading to a decline in CSR performance. Given that an excessive focus on GDP growth often results in a neglect of social accountability and environmental protection, near the end of the term, the governor urges firms to dramatically increase investments in CSR in response to the performance assessment. For instance, the governor may enforce strict environmental regulations or offer more subsidies or tax benefits to firms that perform well in CSR. As a result, firms' CSR performance rises rapidly after the turning point.

The results supported H4, predicting that the U-shaped relationship between a governor's tenure and firms' CSR performance will flatten as the region's marketization level increases.

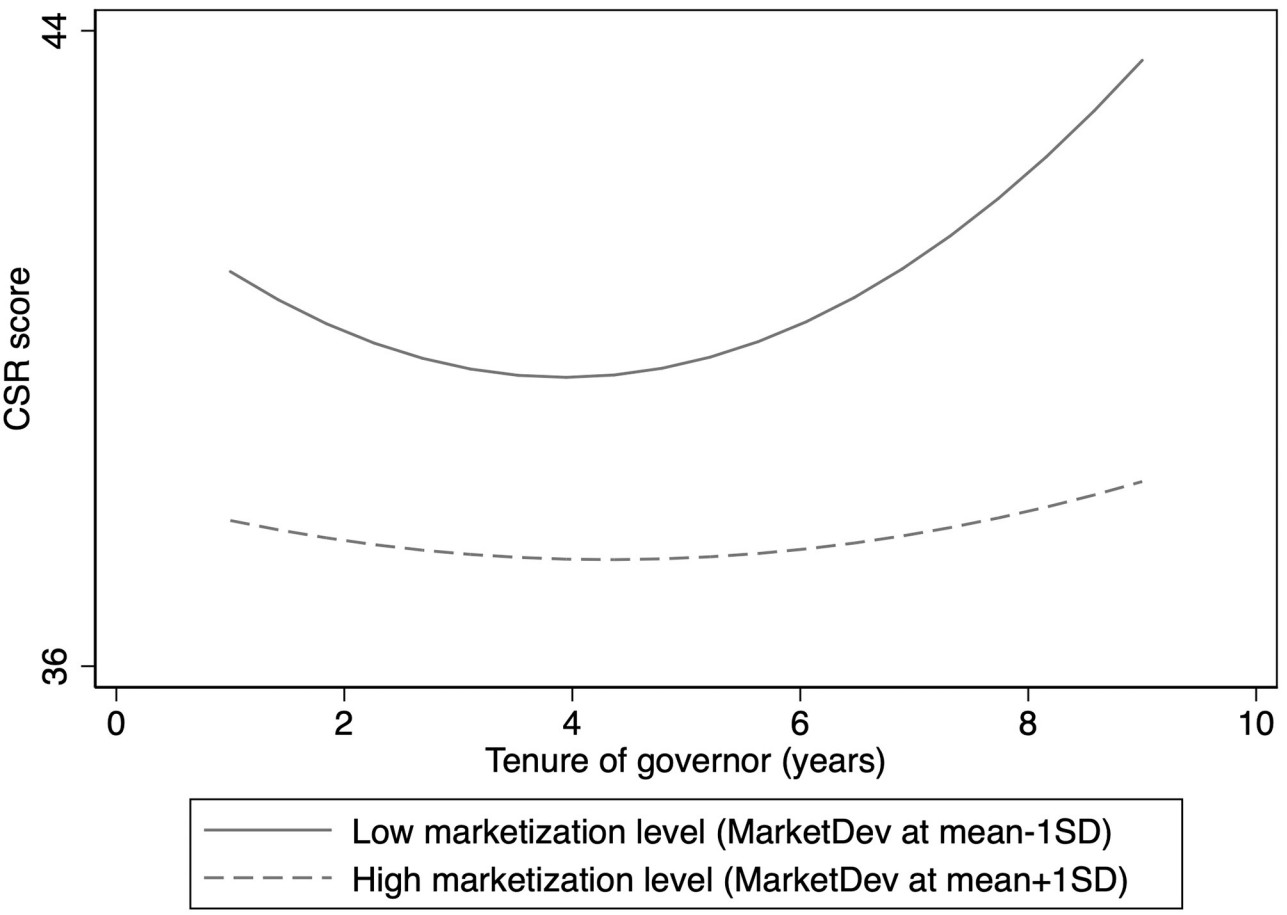

**Fig 2. Moderating effect of regional market development.**

Model 7 in Table 4 shows that the interaction of the market development with the squared term of the governor's tenure is negative and significant at the 10% level. The turning points of the estimated U-curves are calculated and found to be 3.93 and 4.34 under conditions of a low and high marketization level, respectively. These results indicate that the U-shaped effect of the government leader's tenure on CSR will be weakened with the increase of marketization level. Namely, a good institutional environment helps mitigate the negative effect of the governor's tenure on CSR. To visualize the moderating effect of regional market development, I plotted Fig 2. In Fig 2, the low marketization level was one standard deviation (SD) below the mean, and the high marketization level was one SD above the mean. As illustrated in Fig 2, both the downward and upward sides of the relationship flatten in the case of a high marketization level. This indicates that in more market-oriented regions, firms are less dependent on government resources, and the effect of officials' promotion incentives on firm behavior is relatively weak. By contrast, in less market-oriented regions, the government still exercises considerable control over firms, which further strengthens the effect of officials' political incentives on CSR.

### Robustness checks

Endogeneity problem. Although I have used the Heckman two-stage model to mitigate potential sample selection bias, the model may also have reverse causality. On the one hand, governors' tenure can affect firms' CSR performance under their jurisdiction. On the other hand,

provinces with good CSR performance may provide governors with stable tenure. Therefore, I use an instrumental variable (IV) approach to address the endogeneity issue. I choose two instrumental variables: *AbnDismissal* and *AveTenure*. *AbnDismissal* is a dummy variable equal to 1 if the governor is removed from his/her position for a serious disciplinary offense while in office and 0 otherwise. Disciplinary offense (e.g., corruption) directly results in the termination of an official's political career, leading to a break in their term of office. Thus, the abnormal departure is significantly correlated with the official's tenure. However, it has not been documented that the abnormal departure of officials is directly related to the firm's CSR. *AveTenure* refers to the annual average of governors' tenure, which is closely associated with the focal governor's tenure yet not directly related to firms' CSR. I used two-stage least squares (2SLS) regression and generalized method of moments (GMM) regression.

The results are reported in Table 5. The Kleibergen–Paap LM statistics are significant at the 1% level. Namely, the instrumental variables have a sufficient correlation with endogenous variables and reject the null hypothesis that instrumental variables are insufficiently recognized. The Cragg–Donald Wald F statistics reject the null hypothesis that instrumental variables are weakly recognized. The Hansen J statistics are 0.133, which suggests that the instrumental variables cannot be rejected as a null hypothesis of excessive recognition at the 10% level, which indicates that the instrumental variables are exogenous. Therefore, the instrumental variables selected are reasonable. Table 5 shows that the significance of the core variables remains the same, which verifies the empirical rationality of this study.

Moreover, this study uses Lewbel's heteroskedasticity-based approach to the identification [73], which shows that identification can be achieved without imposing any exclusion restrictions if there is a vector Z of exogenous variables and the errors are heteroscedastic. The Z vector can be a subset of the exogenous X vector included in the regression or even Z = X. In the first stage, each endogenous variable is regressed on the Z vector, and the vector of residuals $\hat{\varepsilon}$ is retrieved. These estimated residuals are then used to construct instruments $(Z - \bar{Z})\,\hat{\varepsilon}$, where $\bar{Z}$ is the mean of Z. This approach has been used in the CSR literature to address endogeneity [74]. Specifically, in the first stage, this paper first confirmed the presence of heteroskedasticity in an untabulated analysis. Then, Eq (2) is estimated IV with $(Z - \bar{Z})\,\hat{\varepsilon}$ in the second stage. The regression results in Model 3 of Table 5 reconfirm the U-shaped relationship between the governor's tenure and CSR.

Furthermore, I re-examined the main hypotheses, excluding the samples of municipalities. Although the leaders of municipalities directly under the Central Government are at the provincial ministerial level, unlike other provinces, municipalities directly under the Central Government have certain political and economic peculiarities [62, 75, 76]. The results in Table 6 are generally similar to the previous analysis. In addition, I used CSR Rank—that has a total of 19 levels, and the higher the level, the better the CSR performance—as an alternative measurement of CSR performance and found similar results. Finally, considering the correlation among firms located in the same province, I re-examined the main hypotheses based on the standard errors clustered by provinces and found similar results (See S4 Table).

## Heterogeneity analysis

**Sub-dimensions of CSR.** To further understand provincial top leaders' impact on firms' CSR performance, I re-examined four dimensions of the social responsibility scores: macrocosm, content, technique, and industry. Macrocosm involves the effectiveness of a firm's strategy and corporate governance and evaluation of stakeholders. Content focuses on a firm's social responsibility in the community, environment, product strategy, and sustainable social development. Technique emphasizes the relative balance and related innovation capabilities of

**Table 5. Endogenous test results.**

| Model No. | Model 1 | Model 2 | Model 3 |
|---|---|---|---|
| | IV-2SLS | IV-GMM | Lewbel's (2012) |
| Tenure of governor | -3.747*** | -3.257*** | -1.176*** |
| | (-3.390) | (-3.084) | (-2.749) |
| Tenure of governor$^2$ | 0.380*** | 0.334*** | 0.111** |
| | (3.225) | (2.936) | (2.356) |
| Tenure of party secretary | 0.345* | 0.308 | 0.229 |
| | (1.734) | (1.558) | (1.214) |
| Tenure of party secretary$^2$ | -0.021 | -0.020 | -0.019 |
| | (-0.899) | (-0.868) | (-0.841) |
| Slack resource | 11.544*** | 11.356*** | 10.713*** |
| | (3.822) | (3.763) | (3.577) |
| ROA | -2.113 | -2.279 | -1.797 |
| | (-1.168) | (-1.262) | (-0.986) |
| Leverage | -5.673*** | -5.884*** | -5.996*** |
| | (-3.302) | (-3.436) | (-3.520) |
| Firm age | -0.023 | -0.012 | -0.023 |
| | (-0.394) | (-0.210) | (-0.393) |
| Firm size | 4.207*** | 4.276*** | 4.205*** |
| | (11.070) | (11.336) | (11.075) |
| Foreign income | 1.083 | 1.300 | 1.097 |
| | (0.758) | (0.914) | (0.769) |
| SOE | 1.086 | 1.074 | 1.056 |
| | (1.484) | (1.468) | (1.448) |
| Equity concentration | 0.066*** | 0.068*** | 0.066*** |
| | (2.987) | (3.052) | (2.989) |
| Board size | 0.183** | 0.179* | 0.176* |
| | (1.974) | (1.936) | (1.907) |
| Board independence | 0.287 | 0.057 | -0.358 |
| | (0.100) | (0.020) | (-0.126) |
| Female executive | 1.042 | 1.047 | 1.139 |
| | (1.436) | (1.443) | (1.573) |
| Executive age | 0.307*** | 0.318*** | 0.304*** |
| | (3.132) | (3.250) | (3.117) |
| GDP per capita | 1.406 | 1.587* | 1.365 |
| | (1.463) | (1.664) | (1.427) |
| Population growth | 0.388*** | 0.401*** | 0.313** |
| | (2.901) | (3.002) | (2.382) |
| Fiscal revenue | 1.575*** | 1.561*** | 1.581*** |
| | (2.743) | (2.720) | (2.746) |
| Industry-level CSR | 0.113*** | 0.107*** | 0.108*** |
| | (8.744) | (8.657) | (9.080) |
| Age of governor | 0.229*** | 0.217*** | 0.172*** |
| | (3.957) | (3.791) | (3.305) |
| Age of party secretary | -0.216*** | -0.217*** | -0.211*** |
| | (-3.255) | (-3.270) | (-3.184) |
| Education of governor | 0.726* | 0.735* | 1.128*** |
| | (1.775) | (1.797) | (2.957) |

(*Continued*)

**Table 5.** (Continued)

| Model No. | Model 1 | Model 2 | Model 3 |
|---|---|---|---|
| | IV-2SLS | IV-GMM | Lewbel's (2012) |
| Education of party secretary | 0.304 | 0.246 | 0.266 |
| | (1.097) | (0.898) | (0.984) |
| Birthplace of governor | -1.106* | -1.179* | -1.153* |
| | (-1.821) | (-1.946) | (-1.893) |
| Birthplace of party secretary | -1.977 | -2.017 | -2.197* |
| | (-1.511) | (-1.542) | (-1.701) |
| IMR | 0.911*** | 0.904*** | 0.897*** |
| | (4.514) | (4.481) | (4.455) |
| Constant | -125.057*** | -128.549*** | -125.519*** |
| | (-11.910) | (-12.554) | (-12.074) |
| Observations | 6319 | 6319 | 6319 |
| R-squared | 0.307 | 0.313 | 0.331 |
| F | 39.373 | 39.615 | 38.050 |
| Kleibergen-Paap rk LM | 181.274*** | 181.274*** | 437.898*** |
| Cragg-Donald Wald F | 122.449*** | 122.449*** | 99.073*** |
| Hansen J | 0.133 | 0.133 | 0.017 |

Note. Robust z-statistics (in parentheses) are based on the standard errors clustered by firms to address potential serial correlations in the residuals.

*** $p < 0.01$,

** $p < 0.05$,

* $p < 0.1$.

content. Industry reflects whether the firm is following specific standards in its own industry. The results are presented in Table 7. I found a significant U-shaped relationship between the governor's tenure and the macrocosm, content, and technique dimensions of CSR. This indicates that governors primarily influence a firm's CSR performance in corporate strategy (including the community, environment, and product strategy), governance, and technological innovation.

**Firm's political connections.**   Institutional research suggests that organizational attributes can channel institutional pressures [55, 77]. Some organizational attributes, such as political connections, may make firms vulnerable to officials' influence [78]. Hence, I examine two subsamples separately: politically connected firms whose chairman or CEO has held or currently holds a government position and non-politically connected firms. The results are presented in Table 8. I find a significant U-shaped relationship between the governor's tenure and CSR for firms without political connections, yet does not for those with political connections. This finding shows that non-politically connected firms are more vulnerable to officials' influence, indicating that political connections are likely to provide a "buffer" for the organization from government intervention [79–81].

**Personal characteristics of government leaders.**   Our argument for the U-shaped effect of government leaders' tenure on firms' CSR is based on the premise that officials, driven by their career motivation, must weigh economic growth goals and CSR during their tenure to enhance the prospect of promotion. Studies suggest that the personal characteristics of officials can influence their career motives, thus affecting their trade-offs between promoting economic growth and improving CSR performance. I examined the age, educational background, and current occupation in the central government of officials. The results are presented in Table 9.

**Table 6. Robustness checks excluding the samples of municipalities.**

| Model No. | Model 1 | Model 2 | Model 3 | Model 4 |
|---|---|---|---|---|
| Tenure of governor | -0.908*** | -0.137 | -0.836*** | -0.301 |
| | (-3.312) | (-0.354) | (-2.981) | (-0.765) |
| Tenure of governor$^2$ (H1) | 0.113*** | 0.039 | 0.103*** | 0.061 |
| | (3.132) | (0.902) | (2.751) | (1.360) |
| Tenure of party secretary | 0.150 | 0.171 | 0.162 | 0.194 |
| | (0.529) | (0.604) | (0.566) | (0.676) |
| Tenure of party secretary$^2$ (H2) | 0.016 | 0.009 | 0.014 | 0.000 |
| | (0.412) | (0.244) | (0.342) | (0.003) |
| GDP priority | | -1.410** | | -2.321*** |
| | | (-1.984) | | (-3.020) |
| Tenure of governor*GDP priority | | -1.255** | | -0.591 |
| | | (-2.533) | | (-1.085) |
| Tenure of governor$^2$*GDP priority (H3) | | 0.142** | | 0.053 |
| | | (2.400) | | (0.802) |
| MarketDev | | | -0.845* | -1.160** |
| | | | (-1.816) | (-2.337) |
| Tenure of governor*MarketDev | | | 0.333** | 0.413** |
| | | | (2.502) | (2.560) |
| Tenure of governor$^2$*MarketDev (H4) | | | -0.041** | -0.056** |
| | | | (-2.104) | (-2.345) |
| Slack resource | 9.892*** | 9.856*** | 10.120*** | 10.060*** |
| | (3.187) | (3.163) | (3.258) | (3.250) |
| ROA | -0.413 | -0.460 | -0.373 | -0.408 |
| | (-0.196) | (-0.219) | (-0.178) | (-0.195) |
| Leverage | -5.447*** | -5.379*** | -5.440*** | -5.319*** |
| | (-2.924) | (-2.885) | (-2.933) | (-2.869) |
| Firm age | -0.066 | -0.069 | -0.066 | -0.071 |
| | (-0.954) | (-1.001) | (-0.962) | (-1.028) |
| Firm size | 3.800*** | 3.779*** | 3.783*** | 3.748*** |
| | (8.127) | (8.064) | (8.124) | (8.033) |
| Foreign income | 0.967 | 0.932 | 1.006 | 0.979 |
| | (0.633) | (0.613) | (0.658) | (0.645) |
| SOE | 0.980 | 1.021 | 0.854 | 0.863 |
| | (1.283) | (1.333) | (1.110) | (1.125) |
| Equity concentration | 0.058** | 0.057** | 0.058** | 0.056** |
| | (2.307) | (2.253) | (2.316) | (2.244) |
| Board size | 0.257** | 0.253** | 0.254** | 0.248** |
| | (2.469) | (2.431) | (2.450) | (2.392) |
| Board independence | -3.156 | -2.962 | -3.435 | -3.273 |
| | (-1.131) | (-1.067) | (-1.239) | (-1.191) |
| Female executive | 1.334* | 1.310* | 1.287* | 1.242* |
| | (1.773) | (1.742) | (1.715) | (1.658) |
| Executive age | 0.143 | 0.147 | 0.158 | 0.170 |
| | (1.281) | (1.315) | (1.410) | (1.521) |
| GDP per capita | -3.267*** | -3.248*** | -1.384 | -0.725 |
| | (-2.626) | (-2.596) | (-0.821) | (-0.421) |
| Population growth | 0.180 | 0.153 | 0.210 | 0.194 |

(*Continued*)

**Table 6.** (Continued)

| Model No. | Model 1 | Model 2 | Model 3 | Model 4 |
|---|---|---|---|---|
| | (1.230) | (1.034) | (1.459) | (1.345) |
| Fiscal revenue | 1.857*** | 1.604** | 2.910*** | 2.938*** |
| | (2.880) | (2.500) | (3.500) | (3.488) |
| Industry-level CSR | 0.031* | 0.032* | 0.034* | 0.036** |
| | (1.791) | (1.839) | (1.942) | (2.065) |
| Age of governor | 0.011 | -0.012 | -0.004 | -0.028 |
| | (0.200) | (-0.213) | (-0.072) | (-0.471) |
| Age of party secretary | -0.306*** | -0.280*** | -0.263*** | -0.214*** |
| | (-3.815) | (-3.460) | (-3.447) | (-2.777) |
| Education of governor | 0.538 | 0.371 | 0.400 | 0.193 |
| | (1.349) | (0.967) | (0.962) | (0.478) |
| Education of party secretary | -0.174 | -0.145 | -0.381 | -0.323 |
| | (-0.527) | (-0.454) | (-1.131) | (-0.989) |
| Birthplace of governor | 0.300 | -0.067 | 0.191 | -0.433 |
| | (0.518) | (-0.115) | (0.326) | (-0.733) |
| Birthplace of party secretary | -1.914 | -1.435 | -2.540* | -2.013 |
| | (-1.420) | (-1.078) | (-1.837) | (-1.485) |
| IMR | 0.576** | 0.580** | 0.600*** | 0.612*** |
| | (2.494) | (2.518) | (2.603) | (2.663) |
| Constant | -41.906** | -37.383** | -82.837*** | -89.828*** |
| | (-2.307) | (-2.045) | (-3.018) | (-3.198) |
| Industry FE | yes | yes | yes | yes |
| Year FE | yes | yes | yes | yes |
| Observations | 4540 | 4540 | 4540 | 4540 |
| R-squared | 0.347 | 0.349 | 0.350 | 0.353 |
| F | 8.002 | 7.264 | 7.455 | 6.983 |

Note. Robust t-statistics (in parentheses) are based on the standard errors clustered by firm to address potential serial correlations in the residuals.

*** $p < 0.01$,

** $p < 0.05$,

* $p < 0.1$.

Age limit. Under China's cadre management system, local officials are ineligible for promotion to the next rank once they reach a certain age [82]. Provincial leaders have an official retirement age of 65 [38]. However, they usually are ineligible for promotion at age 63. Literature has suggested that the age limit affects local officials' career concerns [83], influencing their behavior. Therefore, according to the governors' age, I split the sample into two subgroups: terminal officials and promotable officials. Terminal officials are governors who have reached the age of 63 and are currently in their final term, and promotable officials are those under the age of 63. The results (Models 1 and 2 in Table 9) show that the U-shaped relationship between the governor's tenure and CSR is significant, irrespective of age limit.

Educational background. Educational background significantly shapes leaders' thoughts and beliefs, influencing their policy-making and decisions [84]. The academic qualification of officials is also closely related to their ruling characteristics. Generally speaking, the higher the level of education is, the higher the likelihood of promotion. Therefore, I predict that the

**Table 7. The effect of government leaders' tenure on CSR sub-dimensions.**

| Model No. | Model 1 | Model 2 | Model 3 | Model 4 |
|---|---|---|---|---|
| | Macrocosm | Content | Technique | Industry |
| Dependent variables | CSR_M | CSR_C | CSR_T | CSR_I |
| Tenure of governor | -0.236** | -0.304** | -0.129*** | -0.054 |
| | (-2.530) | (-2.422) | (-3.466) | (-1.275) |
| Tenure of governor$^2$ | 0.027** | 0.039** | 0.014*** | 0.007 |
| | (2.246) | (2.461) | (2.925) | (1.398) |
| Tenure of party secretary | 0.003 | 0.115 | 0.047 | -0.050 |
| | (0.045) | (1.140) | (1.505) | (-1.550) |
| Tenure of party secretary$^2$ | 0.012 | -0.004 | -0.001 | 0.007* |
| | (1.422) | (-0.349) | (-0.325) | (1.716) |
| Slack resource | 2.002* | 5.086*** | 1.528*** | 1.260*** |
| | (1.894) | (3.322) | (3.235) | (3.137) |
| ROA | -0.500 | 0.912 | -0.233 | -0.172 |
| | (-0.728) | (0.960) | (-0.824) | (-0.719) |
| Leverage | -2.028*** | -2.976*** | -0.760*** | -0.458** |
| | (-3.160) | (-3.345) | (-2.743) | (-2.059) |
| Firm age | -0.005 | -0.039 | -0.008 | -0.010 |
| | (-0.221) | (-1.256) | (-0.934) | (-1.384) |
| Firm size | 1.343*** | 2.009*** | 0.565*** | 0.434*** |
| | (9.753) | (10.151) | (9.341) | (8.452) |
| Foreign income | 0.336 | 0.571 | 0.191 | -0.129 |
| | (0.621) | (0.738) | (0.910) | (-0.693) |
| SOE | 0.319 | 0.823** | -0.008 | 0.333*** |
| | (1.287) | (2.190) | (-0.077) | (3.666) |
| Equity concentration | 0.024*** | 0.032*** | 0.007** | 0.004 |
| | (3.129) | (2.887) | (2.181) | (1.351) |
| Board size | 0.038 | 0.051 | 0.014 | 0.013 |
| | (1.191) | (1.170) | (0.985) | (1.082) |
| Board independence | 0.026 | -0.900 | -0.025 | 0.058 |
| | (0.026) | (-0.640) | (-0.055) | (0.154) |
| Female executive | 0.420 | 0.642* | 0.142 | 0.036 |
| | (1.569) | (1.789) | (1.202) | (0.337) |
| Executive age | 0.072** | 0.142*** | 0.032** | 0.034*** |
| | (2.059) | (2.759) | (2.122) | (2.681) |
| GDP per capita | -0.001 | 0.223 | 0.129 | 0.080 |
| | (-0.002) | (0.423) | (0.878) | (0.579) |
| Population growth | 0.026 | 0.095 | 0.033* | 0.000 |
| | (0.518) | (1.342) | (1.653) | (0.023) |
| Fiscal revenue | 0.354* | 0.599* | 0.102 | 0.063 |
| | (1.659) | (1.870) | (1.145) | (0.849) |
| Industry-level CSR | 0.014** | 0.015* | 0.001 | 0.000 |
| | (2.565) | (1.790) | (0.331) | (0.161) |
| Age of governor | 0.039** | 0.044* | 0.022*** | 0.006 |
| | (2.178) | (1.670) | (2.787) | (0.812) |
| Age of party secretary | -0.066*** | -0.150*** | -0.020** | -0.027*** |
| | (-2.755) | (-4.343) | (-2.036) | (-2.657) |
| Education of governor | 0.344** | 0.441** | 0.074 | 0.031 |

(*Continued*)

**Table 7.** (Continued)

| Model No. | Model 1 | Model 2 | Model 3 | Model 4 |
|---|---|---|---|---|
| | Macrocosm | Content | Technique | Industry |
| **Dependent variables** | CSR_M | CSR_C | CSR_T | CSR_I |
| | (2.529) | (2.237) | (1.337) | (0.560) |
| Education of party secretary | -0.096 | -0.019 | -0.015 | -0.074 |
| | (-1.050) | (-0.147) | (-0.379) | (-1.421) |
| Birthplace of governor | 0.002 | -0.360 | -0.056 | 0.023 |
| | (0.009) | (-1.134) | (-0.601) | (0.264) |
| Birthplace of party secretary | -0.888* | -1.553** | -0.378* | -0.061 |
| | (-1.791) | (-2.440) | (-1.886) | (-0.421) |
| IMR | 0.325*** | 0.224** | 0.101*** | 0.019 |
| | (4.442) | (2.012) | (3.207) | (0.729) |
| Constant | -28.497*** | -46.184*** | -11.334*** | -10.550*** |
| | (-5.483) | (-6.213) | (-4.826) | (-4.892) |
| Industry FE | yes | yes | Yes | yes |
| Year FE | yes | yes | Yes | yes |
| Observations | 6031 | 6031 | 6031 | 5467 |
| R-squared | 0.391 | 0.287 | 0.377 | 0.286 |
| F | 9.361 | 12.943 | 7.729 | 9.121 |

Note. Given RKS began disclose the scores of CSR in macrocosm, content, and technique sub-dimensions in 2010, the regression models in CSR_M, CSR_C, and CSR_T include 6031 firm-year observation samples covering 2010 to 2019. The score of industry sub-dimension was disclosed from 2011, the model in CSR_I include 5467 firm-year observation samples from 2011 to 2019. Robust t-statistics (in parentheses) are based on the standard errors clustered by firms to address potential serial correlations in the residuals.

*** $p < 0.01$,

** $p < 0.05$,

* $p < 0.1$.

tenure of governors with higher education levels has a stronger impact on CSR than those with lower education levels. I conducted separate tests for the two groups of samples with education levels of master's or above and bachelor's or below. The results (Models 3 and 4 in Table 9) are consistent with our expectations.

Current occupation in the central government. The current occupation of provincial governors in the central government is crucial, as it indicates their remaining promotion space [85]. In China, the ideal promotion path for provincial officials typically moves from alternate member of the Central Committee (ACC) to full member of the Central Committee (FCC), to member of the Political Bureau of the Central Committee (PBCC), and member of the Standing Committee of the Political Bureau of the Central Committee (SCPBCC). Obviously, the higher the position held in the central government, the shorter the remaining promotion space of the governor. I examined samples of governors serving as ACC, FCC, and not serving in the central government separately. Notably, in the sample data, no governor serves on PBCC and SCPBCC. The results (Models 5 to 7 in Table 9) indicate that governors who serve in the central government have a stronger impact on a firm's CSR than those who do not. This finding suggests that given the shorter space for promotion, governors serving in the central government may pay more attention to economic growth, which has a significant negative impact on CSR.

**Table 8. The effect of firms' political connections.**

| Model No. | Model 1 | Model 2 |
|---|---|---|
| | **Political connected firms** | **Non-Political connected firms** |
| Tenure of governor | -0.544 | -0.787** |
| | (-1.278) | (-2.535) |
| Tenure of governor$^2$ | 0.063 | 0.099** |
| | (1.279) | (2.276) |
| Tenure of party secretary | -0.081 | 0.121 |
| | (-0.266) | (0.468) |
| Tenure of party secretary$^2$ | 0.019 | 0.023 |
| | (0.522) | (0.777) |
| Slack resource | 11.208*** | 7.831** |
| | (3.000) | (2.122) |
| ROA | -1.494 | 1.090 |
| | (-0.546) | (0.492) |
| Leverage | -4.899* | -6.644*** |
| | (-1.877) | (-3.132) |
| Firm age | -0.149 | -0.000 |
| | (-1.641) | (-0.002) |
| Firm size | 4.616*** | 3.854*** |
| | (8.224) | (8.349) |
| Foreign income | 1.272 | 0.842 |
| | (0.616) | (0.497) |
| SOE | 1.703* | 1.111 |
| | (1.799) | (1.175) |
| Equity concentration | 0.057 | 0.072*** |
| | (1.642) | (2.823) |
| Board size | 0.115 | 0.108 |
| | (0.968) | (0.959) |
| Board independence | 1.989 | -3.566 |
| | (0.495) | (-1.060) |
| Female executive | 2.312** | 0.407 |
| | (2.283) | (0.422) |
| Executive age | 0.190 | 0.323*** |
| | (1.377) | (2.594) |
| GDP per capita | 1.573 | -1.103 |
| | (1.022) | (-0.965) |
| Population growth | 0.323 | -0.073 |
| | (1.432) | (-0.515) |
| Fiscal revenue | 0.825 | 1.422** |
| | (0.694) | (2.430) |
| Industry-level CSR | 0.042* | 0.022 |
| | (1.804) | (1.010) |
| Age of governor | 0.072 | 0.107 |
| | (0.852) | (1.628) |
| Age of party secretary | -0.242** | -0.265*** |
| | (-2.427) | (-3.315) |
| Education of governor | 1.190* | 0.498 |
| | (1.843) | (1.163) |

(*Continued*)

**Table 8.** (Continued)

| Model No. | Model 1 | Model 2 |
|---|---|---|
| | **Political connected firms** | **Non-Political connected firms** |
| Education of party secretary | -0.222 | -0.323 |
| | (-0.567) | (-0.997) |
| Birthplace of governor | 0.054 | -0.899 |
| | (0.062) | (-1.295) |
| Birthplace of party secretary | 0.153 | -3.683** |
| | (0.077) | (-2.296) |
| IMR | 0.919*** | 0.388 |
| | (3.039) | (1.507) |
| Constant | -108.963*** | -71.111*** |
| | (-5.130) | (-4.172) |
| Industry FE | yes | yes |
| Year FE | yes | yes |
| Observations | 2573 | 3746 |
| R-squared | 0.409 | 0.360 |
| F | 8.474 | 8.347 |

Note. Robust t-statistics (in parentheses) are based on the standard errors clustered by firms to address potential serial correlations in the residuals.

*** $p < 0.01$,

** $p < 0.05$,

* $p < 0.1$.

## Discussion

This paper offers a theoretical analysis of, along with relevant empirical evidence concerning, the effects of local officials' tenure on the CSR performance of firms in the jurisdiction to illustrate the overall effect of political incentives on corporate social behavior.

First, the hypothesis regarding the U-shaped effect of local government officials' tenure on firms' CSR (H1) is verified, which clarifies the ambiguous relationship between political incentives and corporate social behavior to some extent. The empirical evidence so far has been mixed. Some studies have found that political incentives can drive firms to engage in more socially oriented organizational practices [12, 13]. However, it has been argued that political incentives make firms more short-sighted and irresponsible [14]. This paper's results suggest that political incentives' effects on CSR potentially depend on local officials' tenure length. Specifically, political incentives have a negative (positive) impact on firms' CSR before (after) the tenure length of the government leader reaches the turning point (i.e., 4.10 years) of the U-shaped curve. Unlike the existing literature focusing on political events, this paper better captures the overall effect of political incentives by investigating the role of tenure.

Second, this paper analyzes the two boundary conditions of political incentives and CSR. One is the priority of regional GDP growth (H3), which may foster local politicians' short-sighted behavior. Prior research [49] suggests that the high priority given to GDP growth by the local government may undermine firms' actual CSR performance, given China's institutional complexity. It is further proven in this paper that the obsession with GDP growth strengthens the negative effect of the government leader's tenure on CSR. Another boundary condition is regional market development (H4), which is generally considered a critical factor

**Table 9. The effect of governors' personal characteristics.**

| Model No. | Model 1 | Model 2 | Model 3 | Model 4 | Model 5 | Model 6 | Model 7 |
|---|---|---|---|---|---|---|---|
| | Age limit | | Education | | Current occupation in the central government | | |
| | Promotable officials (Age < 63) | Terminal officials (Age >= 63) | Bachelors or below | Master or above | Alternative members of the Central Committee | Full members of the Central Committee | Currently not serving in the central government |
| Tenure of governor | -0.746** | -3.027* | 1.056 | -0.925*** | -1.742 | -0.732** | 0.973 |
| | (-2.447) | (-1.653) | (1.164) | (-3.337) | (-1.622) | (-2.267) | (0.609) |
| Tenure of governor$^2$ | 0.094** | 0.343** | -0.142 | 0.102*** | 0.378* | 0.107*** | -0.367 |
| | (2.086) | (2.417) | (-1.052) | (3.064) | (1.757) | (2.951) | (-1.240) |
| Tenure of party secretary | 0.050 | 1.068 | -0.075 | 0.167 | 1.715*** | -0.195 | 1.765 |
| | (0.181) | (0.895) | (-0.186) | (0.569) | (2.679) | (-0.665) | (1.237) |
| Tenure of party secretary$^2$ | 0.021 | -0.129 | 0.057 | 0.041 | -0.165** | 0.055* | -0.225 |
| | (0.558) | (-0.953) | (1.312) | (1.011) | (-2.391) | (1.732) | (-1.060) |
| Slack resource | 11.322*** | 1.036 | 7.183 | 9.795*** | 8.698** | 8.160** | 18.114*** |
| | (3.800) | (0.180) | (1.282) | (3.105) | (2.172) | (2.170) | (2.792) |
| ROA | 0.004 | -0.965 | -6.432* | 1.317 | -5.064* | 0.602 | 3.767 |
| | (0.002) | (-0.230) | (-1.744) | (0.656) | (-1.822) | (0.273) | (0.854) |
| Leverage | -5.347*** | -7.322** | -10.117*** | -4.796** | -6.324*** | -6.556*** | -1.114 |
| | (-2.990) | (-2.204) | (-3.319) | (-2.519) | (-2.908) | (-3.370) | (-0.286) |
| Firm age | -0.050 | -0.081 | -0.023 | -0.063 | -0.079 | -0.044 | -0.207* |
| | (-0.810) | (-0.775) | (-0.190) | (-1.001) | (-0.897) | (-0.638) | (-1.694) |
| Firm size | 3.922*** | 4.791*** | 4.082*** | 4.066*** | 3.400*** | 4.442*** | 3.570*** |
| | (9.871) | (7.619) | (6.533) | (9.663) | (6.254) | (10.521) | (4.425) |
| Foreign income | 1.031 | 1.100 | 1.125 | 0.749 | 2.245 | 0.757 | 0.069 |
| | (0.708) | (0.466) | (0.428) | (0.484) | (1.128) | (0.465) | (0.018) |
| SOE | 1.465** | 1.101 | -0.559 | 1.772** | 0.911 | 1.309* | 2.786* |
| | (2.017) | (0.944) | (-0.445) | (2.353) | (0.968) | (1.705) | (1.744) |
| Equity concentration | 0.075*** | 0.032 | 0.048 | 0.076*** | 0.094*** | 0.055** | 0.072 |
| | (3.419) | (0.838) | (1.254) | (3.281) | (3.401) | (2.198) | (1.583) |
| Board size | 0.119 | 0.132 | 0.198 | 0.101 | 0.269* | 0.126 | -0.089 |
| | (1.319) | (0.724) | (1.218) | (1.086) | (1.924) | (1.173) | (-0.544) |
| Board independence | -1.584 | -0.224 | -2.410 | -1.509 | -2.337 | 0.321 | -11.614* |
| | (-0.581) | (-0.037) | (-0.483) | (-0.527) | (-0.613) | (0.100) | (-1.889) |
| Female executive | 1.182 | 2.514 | 0.918 | 1.346* | 0.641 | 1.253 | 3.093* |
| | (1.534) | (1.357) | (0.624) | (1.739) | (0.658) | (1.457) | (1.741) |
| Executive age | 0.221** | 0.427** | 0.337** | 0.232** | -0.108 | 0.279*** | 0.385 |
| | (2.196) | (2.430) | (2.144) | (2.169) | (-0.748) | (2.704) | (1.588) |
| GDP per capita | 0.076 | -0.135 | -1.432 | 1.342 | -4.144*** | -0.301 | 5.083** |
| | (0.075) | (-0.050) | (-0.692) | (1.207) | (-2.676) | (-0.276) | (2.075) |
| Population growth | 0.084 | 0.812 | -0.454* | 0.325** | 0.053 | 0.093 | 1.280*** |
| | (0.609) | (1.347) | (-1.798) | (2.111) | (0.278) | (0.611) | (2.706) |
| Fiscal revenue | 1.056* | 4.109*** | 0.874 | 0.940 | 2.306*** | 1.317** | -0.380 |
| | (1.659) | (2.882) | (0.513) | (1.428) | (2.974) | (2.070) | (-0.227) |

(*Continued*)

**Table 9.** (Continued)

| Model No. | Model 1 | Model 2 | Model 3 | Model 4 | Model 5 | Model 6 | Model 7 |
|---|---|---|---|---|---|---|---|
| | **Age limit** | | **Education** | | **Current occupation in the central government** | | |
| | Promotable officials (Age < 63) | Terminal officials (Age >= 63) | Bachelors or below | Master or above | Alternative members of the Central Committee | Full members of the Central Committee | Currently not serving in the central government |
| Industry-level CSR | 0.032* | 0.014 | 0.020 | 0.038* | 0.077*** | 0.029 | -0.047 |
| | (1.677) | (0.392) | (0.647) | (1.934) | (2.919) | (1.420) | (-0.772) |
| Age of governor | 0.091 | -2.182* | -0.217 | 0.220*** | -0.108 | 0.024 | 0.220 |
| | (1.571) | (-1.752) | (-1.350) | (3.480) | (-0.910) | (0.375) | (0.585) |
| Age of party secretary | -0.237*** | 0.033 | -0.165 | -0.244*** | -0.447*** | -0.277*** | -0.387* |
| | (-3.523) | (0.145) | (-0.748) | (-3.399) | (-2.955) | (-3.837) | (-1.753) |
| Education of governor | 0.694* | -2.860 | — | 0.639 | 2.052*** | 0.183 | -2.646 |
| | (1.801) | (-1.247) | — | (0.984) | (2.588) | (0.394) | (-1.163) |
| Education of party secretary | -0.488* | 2.450** | 0.246 | 0.228 | -1.171 | -1.111*** | 2.047** |
| | (-1.827) | (2.213) | (0.230) | (0.763) | (-1.450) | (-3.001) | (2.391) |
| Birthplace of governor | -0.165 | -1.036 | 0.191 | -1.260 | -0.016 | -0.544 | -3.844 |
| | (-0.279) | (-0.432) | (0.110) | (-1.620) | (-0.015) | (-0.693) | (-1.184) |
| Birthplace of party secretary | -2.274* | — | -0.564 | -6.563*** | 2.869 | -2.396 | -2.745 |
| | (-1.724) | — | (-0.288) | (-4.515) | (1.099) | (-1.435) | (-0.857) |
| IMR | 0.548*** | 1.127*** | 0.838** | 0.601*** | 0.442 | 0.738*** | 0.326 |
| | (2.603) | (3.075) | (2.374) | (2.624) | (1.507) | (3.324) | (0.641) |
| Constant | -78.554*** | -29.011 | -52.706* | -104.529*** | -8.319 | -83.759*** | -104.591** |
| | (-5.304) | (-0.440) | (-1.930) | (-6.698) | (-0.333) | (-5.105) | (-2.407) |
| Industry FE | yes | yes | yes | yes | yes | yes | yes |
| Year FE | yes | yes | yes | yes | yes | yes | yes |
| Observations | 5342 | 977 | 1224 | 5095 | 1457 | 4032 | 830 |
| R-squared | 0.352 | 0.452 | 0.435 | 0.348 | 0.339 | 0.399 | 0.339 |
| F | 11.713 | 7.349 | 4.629 | 12.197 | 6.127 | 10.852 | 4.849 |

Note. Robust t-statistics (in parentheses) are based on the standard errors clustered by firms to address potential serial correlations in the residuals.

*** p<0.01,

** p<0.05,

* p<0.1.

in influencing firms' CSR [2, 86, 87]. Literature suggests that in regions with good market development, a firm's resource dependency on the government is weakened [55]. This paper documents that good market development can mitigate the negative effect of the government leader's tenure on CSR.

Last but not least, this paper also examines the influence of the functional differentiation between party and government officials (H2). Prior research shows that functional differentiation enables government and party officials to differently influence firm behavior, such as corporate diversification [36]. This paper further indicates that local party and government leaders also motivate the CSR activities of firms differently.

## Conclusion

A large body of literature focuses on the firm- and institution-level factors driving CSR, with little attention paid to the impact of political incentives. This study investigates the impact of local officials' political incentives (measured by the tenure) on firms' CSR based on China's A-share-listed companies covering 2009–2019, drawing the following conclusions: (1) there is a significant U-shaped relationship between local government leaders' tenure and the CSR performance of public firms located within their jurisdiction; (2) such a U-shaped effect will be strengthened in regions with a high priority of the regional GDP growth; (3) such a U-shaped effect will be weakened in regions with good market development; (4) no significant empirical evidence suggests that local party leaders' tenure influences firms' CSR. The findings of this paper will help clarify the ambiguous relationship between political incentives and corporate social behavior.

### Theoretical implications

This paper makes contributions in several ways. First, this paper contributes to the growing literature on the determinants of CSR in China. Although the extensive literature has proved that the CSR level of an enterprise is jointly determined by a series of factors, including the firm's external institutional factors [3, 61, 88], its own organizational characteristics [9, 78, 89, 90], internal governance structure [91, 92], managers' experience and values [8, 93, 94], and financial performance [95, 96], little attention has been paid to political incentives for state officials. Unlike Kong et al. (2021), focusing on the positive effects of political promotion events on CSR [12], this paper documents both the positive and negative impact of political incentives on firms' CSR, depending on the tenure length of local officials. The finding in this paper enriches our understanding of the political determinants of CSR in emerging markets.

This study also expands the literature on the role of political promotion incentives in China. Prior studies suggest that state officials' promotion incentives can influence the behavior and performance of firms, including financial performance [97], green innovation [98], labor investment efficiency [99], pollution discharges [14], and stock price behavior [100, 101]. Unlike their primary focus on political promotion or turnover events, this study concentrates on the overall impact of officials' promotion incentives on firm behavior throughout their tenure and how this impact changes, revealing the double-edged sword effect of promotion incentives on CSR. The findings of this paper show that the tenure length and functional differentiation of local officials may be key factors in exploring the ambiguous relationship between political incentives and firm behavior.

This study adds to the understanding of multitasking agency problems for government agents. Career concern theories suggest that top bureaucrats are driven mainly by the outcomes of their mandated tasks [26–29]. Under this framework, officials choose their effort levels and distribution of efforts across tasks to maximize their signaled capability to the market (or the principal). The results of this paper further show that the double-edged sword effect of political incentives on CSR results from government officials weighing economic growth and CSR goals. In addition, such an effect is influenced by the priority of economic growth and market development in regions where they are located. Overall, this study aids us in further understanding how officials balance multiple tasks.

### Practical implications

This paper adds to the understanding of political institutions in China, demonstrating the double-edged sword effect of promotion incentives on firms' CSR behavior. The research in this paper may have policy implications in the following aspects. First, the central government

should appropriately increase the weight of social responsibility indicators in the promotion assessment system for local officials to correct the short-term behavior of local officials who attach importance to short-term economic growth and neglect CSR improvement. Second, the central government should strengthen the supervision of the excessive worship of local governments' GDP growth to prevent local officials from achieving economic growth at the expense of social and environmental sustainability. Third, speeding up the regional marketization process and ensuring the decisive role of the market in resource allocation is also an effective way to curb local officials' short-sighted behavior.

## Limitations and future research

It is essential to recognize the limitations of our study. The sample in this paper comes from listed firms that voluntarily disclose CSR reports and are included in RKS, and it may not be representative of every Chinese firm. I do not know whether our findings apply to companies that are not publicly listed or do not explicitly disclose CSR reports. Future research should use a more extensive and diverse sample or conduct comparative research to validate our findings.

## Supporting information

**S1 Table. RKS MCTI rating system.**
(DOC)

**S2 Table. Correlation metrics.**
(DOC)

**S3 Table. Statistical tests of the U-shaped relationship between the government leader's tenure and CSR.**
(DOC)

**S4 Table. Robustness checks with the standard errors clustered by provinces.**
(DOC)

## Author Contributions

**Conceptualization:** Yunyu Wu.

**Data curation:** Yunyu Wu.

**Formal analysis:** Yunyu Wu.

**Investigation:** Yunyu Wu.

**Methodology:** Yunyu Wu.

**Software:** Yunyu Wu.

**Supervision:** Yunyu Wu.

**Validation:** Yunyu Wu.

**Visualization:** Yunyu Wu.

**Writing – original draft:** Yunyu Wu.

**Writing – review & editing:** Yunyu Wu.

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
