## [Decision Letter · Decision Letter 0]

2 Dec 2022

PONE-D-22-31269Do political incentives promote or inhibit corporate social responsibility? The role of local officials’ tenurePLOS ONE

Dear Dr. Wu,

Thank you for submitting your manuscript to PLOS ONE. After careful consideration, we feel that it has merit but does not fully meet PLOS ONE’s publication criteria as it currently stands. Therefore, we invite you to submit a revised version of the manuscript that addresses the points raised during the review process.

ACADEMIC EDITOR: The reviewers are experts in the area and have prepared a careful and fair review. I do appreciate your efforts in writing the manuscript and I find the topic interesting and worth pursuing. The paper is generally well written, and analyses have been conducted properly. However, there are some places highlighted by reviewers that you should carefully address before publication. Please find all comments in the review reports. I will not repeat them here to avoid your confusions. 

We look forward to receiving your revised manuscript.

Kind regards,

Vu Quang Trinh, PhD

Academic Editor

PLOS ONE

Journal Requirements:

Additional Editor Comments:

NA

Reviewers' comments:

Reviewer's Responses to Questions

**Comments to the Author**

1. Is the manuscript technically sound, and do the data support the conclusions?

Reviewer #1: Yes

Reviewer #2: Yes

2. Has the statistical analysis been performed appropriately and rigorously? 

Reviewer #1: Yes

Reviewer #2: Yes

3. Have the authors made all data underlying the findings in their manuscript fully available?

Reviewer #1: Yes

Reviewer #2: Yes

4. Is the manuscript presented in an intelligible fashion and written in standard English?

Reviewer #1: Yes

Reviewer #2: Yes

5. Review Comments to the Author

Reviewer #1: This is a very interesting article, the analysis is sound, and the results are convincing. It is well embedded in the literature and its contribution is significant. The author applies the Heckman method to eliminate selection bias and instrumental variables to handle further endogeneity concerns. I suggest minor improvements on two points:

1. Present the size effect. Show that the coefficients are not only statistically significant but also economically significant. It might be worthwhile to plot the U-curve as a function of the moderating factors.

2 Explain in more detail how the CSR indicator is calculated. Support that the CSR measure is a good proxy for the latent variable. In this context, consider that firms have an interest in hiding CSR problems or in presenting themselves in a better light than they actually are (greenwashing). The less free the media (e.g. in some rural regions), the greater this type of bias may be. I recommend the following two articles on this issue:

Berlinger, E., Keresztúri, J. L., Lublóy, Á., & Tamásné, Z. V. (2022). Press freedom and operational losses: The monitoring role of the media. Journal of International Financial Markets, Institutions and Money, 77, 101496.

You, J., Zhang, B., & Zhang, L. (2018). Who captures the power of the pen?. The Review of Financial Studies, 31(1), 43-96.

Reviewer #2: Summary

This paper estimates the relationship between the political incentives of local government officials (proxied by their tenure) and corporate social responsibility (CRS) of firms located within their jurisdiction in China. The paper hypothesizes that, under the institutional circumstances in which businesses operate, the relationship between the tenure of local officials and CRS is U-shaped, along with other hypotheses. Using data from 985 firms covering the period 2009-2019, the paper finds a U-shaped relationship between the tenure and CRS performance. By contrast the paper finds no evidence that party officials’ tenure has any effect on firms’ CRS.

General Comments

This is a well written paper which addresses a very important policy-return issue that speaks to literature in many fields The author has painstakingly explained the hypotheses being tested, the methods and the analysis. The author also tries to identify and address the potential methodological challenges with the data and its estimation. In spite of this, there are some important issues to be addressed before the paper can be considered for publication in a journal.

Specific Comments

Conceptually, the link between the political incentives of local officials and corporate social responsibility of firms is not well established in the paper. The author indicates [page 4 lines 78 79] that the local government’s control of critical resources could dramatically affect a firm’s competitive position. However, that actions of the local officials could somehow induce CSR from firms a jump. Even though the author also indicates that the firms’ CSR investment has a weight in the evaluation criteria of local officials, how much is this weight? More generally, the author should provide specific channels through which the tenure of the local officials could influence CRS especially that of private firms.

On the methodology, the decision to restrict the study sample to only firms that engaged in CRS and exclude those that did not (page 16 lines 307-308) is questionable. I would have expected that all firms would be included so that for years in which firms did not engage in CRS, they are marked as zero. The author has employed the Heckman two-stage selection approach to address the resulting sample selection problems inherent in their approach but it seems strange to impose this restrict and use a sample selection to address it. At the very minimum, I expect the author to rerun the analysis where firms that are not engaged in CRS are included to see whether the results will remain the same. Otherwise, the author should provide a strong justification for not using that approach.

Minor comments

Some of the variable names are not self-explanatory. For instance, “nature” is a dummy for being state-owned. It is important for the author to add sufficient notes to the descriptive table, (Table 2) including variable definitions to make the tables self-contained.

Also, it is not too clear how the standard errors are treated in the empirical estimation but is seems that the author may have to adjust for correlation among firms located in the same province/locality.

6. PLOS authors have the option to publish the peer review history of their article (what does this mean?). If published, this will include your full peer review and any attached files.

Reviewer #1: No

Reviewer #2: No

---

## [Author Response · Author response to Decision Letter 0]

16 Jan 2023

Responses to Reviewers’ Comments

SUGGESTIONS FROM ACADEMIC EDITOR:

The reviewers are experts in the area and have prepared a careful and fair review. I do appreciate your efforts in writing the manuscript and I find the topic interesting and worth pursuing. The paper is generally well written, and analyses have been conducted properly. However, there are some places highlighted by reviewers that you should carefully address before publication. Please find all comments in the review reports. I will not repeat them here to avoid your confusions.

COMMENTS TO THE AUTHOR:

Reviewer #1:

This is a very interesting article, the analysis is sound, and the results are convincing. It is well embedded in the literature and its contribution is significant. The author applies the Heckman method to eliminate selection bias and instrumental variables to handle further endogeneity concerns. I suggest minor improvements on two points:

1. Present the size effect. Show that the coefficients are not only statistically significant but also economically significant. It might be worthwhile to plot the U-curve as a function of the moderating factors.

Response:

I am grateful to the reviewer for raising this valuable point. I have now implemented the suggestion made by the reviewer and plotted the U-curve to present the size effect (As shown in Figs 1 and 2). Fig 1 visualizes the moderating effect of regional GDP priority on the U-shaped relationship between the governor’s tenure and firms’ CSR. As shown in Fig 1, both the downward and upward sides of the relationship steepen under conditions of high GDP priority. This indicates that the high priority of GDP growth strengthens the U-shaped effect of the government leader’s tenure on CSR. Specifically, high GDP growth targets spur newly appointed governors to boost GDP quickly using various policy tools (e.g., relaxed environmental regulations), thus leading to a decline in CSR performance. Given that an excessive focus on GDP growth often results in a neglect of social accountability and environmental protection, near the end of the term, the governor urges firms to dramatically increase investments in CSR in response to the performance assessment. For instance, the governor may enforce strict environmental regulations or offer more subsidies or tax benefits to firms that perform well in CSR. As a result, firms’ CSR performance rises rapidly after the turning point. Please see the “Main results” sub-section in the “Results” section for details (page 26, lines 423-439, in the revised version).

Fig 1. Moderating effect of regional GDP priority.

Fig 2 visualizes the moderating effect of regional market development. In Fig 2, the low marketization level was one standard deviation (SD) below the mean, and the high marketization level was one SD above the mean. As illustrated in Fig 2, both the downward and upward sides of the relationship flatten in the case of a high marketization level. In addition, the turning points of the estimated U-curves are calculated and found to be 3.93 and 4.34 under conditions of a low and high marketization level, respectively. These results indicate that the U-shaped effect of the government leader’s tenure on CSR will be weakened with the increase of regional marketization level. More specifically, in more market-oriented regions, firms are less dependent on government resources, and the effect of officials’ promotion incentives on firm behavior is relatively weak. By contrast, in less market-oriented regions, the governments still exercise considerable control over firms, which further strengthens the effect of officials’ political incentives on firms’ CSR. Please see the “Main results” sub-section in the “Results” section for details (page 27, lines 440-455, in the revised version).

Fig 2. Moderating effect of regional market development.

2. Explain in more detail how the CSR indicator is calculated. Support that the CSR measure is a good proxy for the latent variable. In this context, consider that firms have an interest in hiding CSR problems or in presenting themselves in a better light than they actually are (greenwashing). The less free the media (e.g. in some rural regions), the greater this type of bias may be. I recommend the following two articles on this issue:

Berlinger, E., Keresztúri, J. L., Lublóy, Á., & Tamásné, Z. V. (2022). Press freedom and operational losses: The monitoring role of the media. Journal of International Financial Markets, Institutions and Money, 77, 101496.

You, J., Zhang, B., & Zhang, L. (2018). Who captures the power of the pen?. The Review of Financial Studies, 31(1), 43-96.

Response:

I am grateful to the reviewer for raising this valuable point on the CSR indicator. I agree with the reviewer’s observation that companies may hide or overstate their actual CSR performance for some reason (e.g., poor media monitoring), thus leading to the bias of CSR measures. In this study, I used the social responsibility scores provided by Rankins CSR Rating (RKS) to measure a firm’s CSR performance. The CSR evaluation system established by RKS is based on the Global Reporting Initiative (3.0) framework and incorporates Chinese-specific CSR elements. The rating system for RKS is presented in S1 Table. Although far from perfect, the RKS dataset provides a multidimensional assessment of CSR activities conducted by Chinese listed ﬁrms based on a variety of sources of information (e.g., firms’ publicly released social responsibility reports, official websites, and news media) and using consistent and systematic criteria from year to year. In practice, the RKS dataset has been widely used in previous CSR studies in China [1-4] as a measure of firms’ substantive engagement in social activities with satisfactory results. 

In addition, validity tests of this measure were also done by studies [5, 6]. The investigations and results on validity tests show that the RKS score reflects how well a company is actually engaged in CSR, which gives us further confidence in the RKS measure. Hence, it is highly credible to use RKS scores to represent CSR performance. We believe this measure can effectively mitigate the measurement bias caused by companies deliberately hiding or exaggerating CSR reporting. Please see the “Dependent variables” section in the revised manuscript for details (pages 13-14, lines 282-298, in the revised version).

Reviewer #2:

SUMMARY

This paper estimates the relationship between the political incentives of local government officials (proxied by their tenure) and corporate social responsibility (CRS) of firms located within their jurisdiction in China. The paper hypothesizes that, under the institutional circumstances in which businesses operate, the relationship between the tenure of local officials and CRS is U-shaped, along with other hypotheses. Using data from 985 firms covering the period 2009-2019, the paper finds a U-shaped relationship between the tenure and CRS performance. By contrast the paper finds no evidence that party officials’ tenure has any effect on firms’ CRS.

GENERAL COMMENTS

This is a well written paper which addresses a very important policy-return issue that speaks to literature in many fields The author has painstakingly explained the hypotheses being tested, the methods and the analysis. The author also tries to identify and address the potential methodological challenges with the data and its estimation. In spite of this, there are some important issues to be addressed before the paper can be considered for publication in a journal.

SPECIFIC COMMENTS

1. Conceptually, the link between the political incentives of local officials and corporate social responsibility of firms is not well established in the paper. The author indicates [page 4 lines 78 79] that the local government’s control of critical resources could dramatically affect a firm’s competitive position. However, that actions of the local officials could somehow induce CSR from firms a jump. Even though the author also indicates that the firms’ CSR investment has a weight in the evaluation criteria of local officials, how much is this weight? More generally, the author should provide specific channels through which the tenure of the local officials could influence CSR especially that of private firms.

Response:

I am grateful to the reviewer for raising this valuable point. I have now implemented the suggestions made by the reviewer. I have rewritten relevant content on the link between local officials’ political incentives and firms’ CSR, including an analysis of specific channels through which their tenure affects CSR. Here, I briefly answer the reviewers’ questions. Please see the “Introduction ”section in the revised manuscript for details (pages 4-6, lines 75-109, in the revised version).

1.1 How has the link between the political incentives of local officials and firms’ CSR been established? 

Since 2000, the central government has implemented several regulations to incorporate CSR as an essential factor in business operations in China. In 2006, the Shenzhen Stock Exchange and Shanghai Stock Exchange issued “The Social Responsibility Guidelines,” requiring listed companies to establish social responsibility systems and to form social responsibility reports [7]. Subsequently, in 2009, the central government reformed the performance evaluation standards for local officials and added the social responsibility performance of officials’ jurisdiction to the new system. Unlike a single economic performance indicator in the original version, the revised indicator is an aggregate of a group of components, including economic performance, social responsibility, resident welfare, unemployment rate, and relative equity [8]. As a result, the weight of CSR-related indicators remarkably increased while economic indicators experienced a significant drop in evaluation weight [9]. 

1.2 What are the specific channels through which the tenure of local officials could influence firms’ CSR?

Under China’s unique centralized system, local officials have a set of policy tools to impose their influence on firms’ decisions, including CSR. Given the dual goals of economic growth and CSR facing local officials, they may employ different policy tools to achieve their political goals at different stages of their tenure. Early in the tenure, local officials tend to implement a series of policies and regulations to boost rapid economic growth. For instance, they can directly issue commands on production strategies to state-owned enterprises (SOEs) [10]. They can also facilitate the production of private firms by relaxing environmental regulations or providing cheaper land and credit [11]. Doing so leads to a decline in CSR. 

However, as the tenure continues to be extended, local officials urge firms to improve their CSR through various channels. For SOEs, the government has a great deal of authority over the firm’s daily operations and can directly issue commands regarding those firms’ CSR compliance [10]. In particular, in some cases, the government can even temporarily close down some regulated SOEs to meet environmental standards. For private firms, the government can make more frequent inspections and urge them to fulfill CSR, such as by adopting cleaner production methods or installing expensive pollution abatement equipment [12]. Also, the government could offer subsidies or tax benefits to companies that perform well in CSR to encourage CSR investments [1].

2. On the methodology, the decision to restrict the study sample to only firms that engaged in CSR and exclude those that did not (page 16 lines 307-308) is questionable. I would have expected that all firms would be included so that for years in which firms did not engage in CSR, they are marked as zero. The author has employed the Heckman two-stage selection approach to address the resulting sample selection problems inherent in their approach but it seems strange to impose this restrict and use a sample selection to address it. At the very minimum, I expect the author to rerun the analysis where firms that are not engaged in CSR are included to see whether the results will remain the same. Otherwise, the author should provide a strong justification for not using that approach.

Response:

I am grateful to the reviewer for raising this valuable point on the methodology. I agree with the reviewer’s observation that the inclusion of all firms in the study sample helps improve the results’ robustness to some extent. However, this may not be very applicable to the present study. In the following, I elaborate on the reasons for selecting the current study sample and adopting the Heckman two-stage selection approach. 

Before doing so, I first apologize for the inaccurate statement in the previous manuscript. In the previous version, I stated that “… this study sampled firms from those engaged in CSR activities and excluded firms that did not...” (page 16, lines 307-308, in the previous version). This statement may not be rigorous enough. In response, I fixed this in the revised manuscript. The revised statement is that “… this study sampled firms from those disclosed CSR reports and excluded firms that did not…” (page 17, lines 338-340, in the revised version).

Moreover, I believe it is necessary to reinterpret the decision on the current sample selection. This study sampled firms that disclosed CSR reports and were included in Rankins CSR Rating (RKS), as described in the “Sample and data sources” section (page 12, lines 252-253, in the revised version). This means that firms that did not disclose CSR reports were not considered. In other words, this study selected firms that engaged in CSR activities and publicly disclosed CSR information (i.e., CSR reports) and excluded those that did not disclose CSR reports. For firms that do not publicly release CSR reports, there are two possible scenarios for their CSR performance. On the one hand, they may not actually engage in CSR; on the other hand, they may be involved in CSR activities but not publicly announced. Unfortunately, in either case, the CSR performance of these companies could not be determined for the current study.

2.1 Why did the study sample cover firms that disclosed CSR reports and were included in Rankins CSR Rating (RKS) and exclude those that did not disclose CSR reports?

It has been recognized that measuring CSR performance is a difficult task [13, 14] because of both methods and disclosure incentives. Especially for firms that do not release CSR reports, it is extremely challenging to gauge their actual CSR performance. It could be argued that these firms might not be devoting any attention or resources to CSR and therefore chose not to submit the reports. This could be true, but we have no evidence to substantiate this view. However, it is also possible that they have engaged in CSR activities but opt for nondisclosure. After all, in China, the disclosure of CSR reporting is voluntary for most companies, although a small number are required to make it mandatory. In either case, the social performance of these non-disclosing firms cannot be determined by the current study. In this vein, marking the CSR performance of these companies as 0 would cause significant measurement bias, undermining the reliability and validity of the study. 

In contrast, sampling from companies that have disclosed CSR reports better ensures the reliability of the results. In particular, I used the social responsibility scores provided by RKS to measure CSR performance. RKS is a third-party rating agency for CSR in China and one of the major sources for rating the CSR engagement of Chinese firms. It comprehensively evaluates a firm’s CSR performance based on its publicly released social responsibility reports, official websites, and news media. The RKS dataset has been widely used in previous CSR studies on China [1-4] as a measure of firms’ substantive engagement in CSR activities with satisfactory results. In addition, validity tests of this measure show that the social responsibility scores provided by RKS well reflect a firm’s actual CSR performance [5, 6].

2.2 Why use Heckman two-stage selection approach to address sample selection bias?

As mentioned above, sampling from firms that disclosed CSR reports and were included in the RKS dataset can well ensure the reliability of data regarding CSR performance. However, this approach may suffer from nonrandom sampling bias because firms that disclosed CSR reports may differ substantially from those that did not. To correct for such sampling bias, I followed prior studies using a Heckman two-stage model to test our hypotheses [3, 15-17]. Specifically, in the first stage, I used a variety of firm and industrial variables to predict the likelihood of a firm disclosing CSR reporting. A probit model with the entire sample was run, including both firms disclosing CSR reporting and those not disclosing CSR reporting. Based on the results of the probit model, I calculated an adjustment term, the inverse Mill’s ratio (IMR), which I included as a control variable in the main second-stage model to control for the selection bias. In the second stage, we found positive and significant coefficients on IMR (Seen in Table 4). Results indicate that the findings of this study are still robust after controlling for the sampling bias.

In summary, the actual CSR performance of firms that did not disclose CSR reports cannot be determined the present study. Sampling from companies that have disclosed CSR reports better ensures the reliability of the results. 

MINOR COMMENTS

3. Some of the variable names are not self-explanatory. For instance, “nature” is a dummy for being state-owned. It is important for the author to add sufficient notes to the descriptive table, (Table 2) including variable definitions to make the tables self-contained.

Response:

I appreciate the reviewer for raising this valuable point. I agree that some variable names in the previous version were not self-explanatory. I have now implemented the suggestion made by the reviewer and have renamed some variables. Additionally, I have added notes at the bottom of Table 2. Please refer to Tables 1 and 2 in the revised manuscript for details.

4. Also, it is not too clear how the standard errors are treated in the empirical estimation but is seems that the author may have to adjust for correlation among firms located in the same province/locality.

Response:

I am grateful to the reviewer for raising this point. In the empirical estimation, standard errors were clustered at the firm level since we have multiple observations on each firm. Doing so helps solve the heteroskedasticity and autocorrelation problems. I added notes on how the standard errors are treated at the bottom of each table. Also, I agree that there may be a correlation among firms in the same province/locality. I have now implemented the suggestion made by the reviewer and have re-examined the main hypotheses in the “Robustness checks” section based on the standard errors clustered by provinces and found similar results. Resression resuls are presented in S4 Table.

1. Kong D, Cheng X, Jiang X. Effects of political promotion on local firms? Social responsibility in China. Econ Modelling. 2021;95:418-29. doi: 10.1016/j.econmod.2020.03.009.

2. Li S, Lu JW. A dual-agency model of firm CSR in response to institutional pressure: Evidence from Chinese publicly listed firms. Acad Manage J. 2020;63(6):2004-32. doi: 10.5465/amj.2018.0557.

3. Du J, Bai T, Chen S. Integrating corporate social and corporate political strategies: Performance implications and institutional contingencies in China. J Bus Res. 2019;98:299-316. doi: 10.1016/j.jbusres.2019.02.014.

4. Lau C, Lu Y, Liang Q. Corporate social responsibility in China: A corporate governance approach. J Bus Ethics. 2016;136(1):73-87. doi: 10.1007/s10551-014-2513-0.

5. Marquis C, Qian C. Corporate social responsibility reporting in China: Symbol or substance? Organ Sci. 2014;25(1):127-48. doi: 10.1287/orsc.2013.0837.

6. Luo XR, Wang D, Zhang J. Whose call to answer: Institutional complexity and firms’ CSR reporting. Acad Manage J. 2017;60(1):321-44. doi: 10.5465/amj.2014.0847.

7. Chen YC, Hung M, Wang Y. The effect of mandatory CSR disclosure on firm profitability and social externalities: Evidence from China. J Acc Econ. 2018;65(1):169-90. doi: 10.1016/j.jacceco.2017.11.009.

8. CPC TODotCCot. 2012 Nov 21 [cited 2023 Jan 5]. Available from: http://wsqdjw.gov.cn/zcfg/201211/t20121121_745004.html.

9. Wang Z. Reassessing the performance evaluation system in the Xi Jinping era: Changes and implications. East Asia. 2018;35(1):59-77. doi: 10.1007/s12140-018-9281-x.

10. Naughton BJ. The Chinese economy: Transitions and growth: MIT press; 2006.

11. Haggard S, Huang Y. The political economy of private-sector development in China. China’s Great Economic Transformation2008. p. 337-74.

12. He G, Wang S, Zhang B. Watering down environmental regulation in China. Quart J Econ. 2020;135(4):2135-85. doi: 10.1093/qje/qjaa024.

13. Hackston D, Milne MJ. Some determinants of social and environmental disclosures in New Zealand companies. Accounting, Auditing & Accountability Journal. 1996;9(1):77-108. doi: 10.1108/09513579610109987.

14. Krippendorff K. Reliability in content analysis: Some common misconceptions and recommendations. Human Communication Research. 2004;30(3):411-33. doi: 10.1093/hcr/30.3.411.

15. Sun P, Hu HW, Hillman AJ. The dark side of board political capital: Enabling blockholder rent appropriation. Acad Manage J. 2016;59(5):1801-22. doi: 10.5465/amj.2014.0425.

16. Wang H, Qian C. Corporate philanthropy and corporate financial performance: The roles of stakeholder response and political access. Acad Manage J. 2011;54(6):1159-81. doi: 10.5465/amj.2009.0548.

17. Zhang J, Marquis C, Qiao K. Do political connections buffer firms from or bind firms to the government? A study of corporate charitable donations of Chinese firms. Organ Sci. 2016;27(5):1307-24. doi: 10.1287/orsc.2016.1084.

---

## [Decision Letter · Decision Letter 1]

6 Mar 2023

Do political incentives promote or inhibit corporate social responsibility? The role of local officials’ tenure

PONE-D-22-31269R1

Dear Dr. Wu,

We’re pleased to inform you that your manuscript has been judged scientifically suitable for publication and will be formally accepted for publication once it meets all outstanding technical requirements.

Kind regards,

Vu Quang Trinh, PhD

Academic Editor

PLOS ONE

Additional Editor Comments (optional):

All reviewers' comments have been properly addressed and the paper has been significantly improved; therefore, I recommend the acceptance of the paper.

Reviewers' comments:

Reviewer's Responses to Questions

**Comments to the Author**

1. If the authors have adequately addressed your comments raised in a previous round of review and you feel that this manuscript is now acceptable for publication, you may indicate that here to bypass the “Comments to the Author” section, enter your conflict of interest statement in the “Confidential to Editor” section, and submit your "Accept" recommendation.

Reviewer #1: All comments have been addressed

Reviewer #2: All comments have been addressed

2. Is the manuscript technically sound, and do the data support the conclusions?

Reviewer #1: Yes

Reviewer #2: Yes

3. Has the statistical analysis been performed appropriately and rigorously? 

Reviewer #1: Yes

Reviewer #2: Yes

4. Have the authors made all data underlying the findings in their manuscript fully available?

Reviewer #1: Yes

Reviewer #2: Yes

5. Is the manuscript presented in an intelligible fashion and written in standard English?

Reviewer #1: Yes

Reviewer #2: Yes

6. Review Comments to the Author

Reviewer #1: All my comments have been properly addressed and the paper has been significantly improved, therefore, I recommend the acceptance of the paper. I congratulate the authors(s).

Reviewer #2: The author(s) have carefully and dutifully addressed all my comments. I do not have any additional comments.

7. PLOS authors have the option to publish the peer review history of their article (what does this mean?). If published, this will include your full peer review and any attached files.

Reviewer #1: No

Reviewer #2: No

---

## [Editor Report · Acceptance letter]

8 Mar 2023

PONE-D-22-31269R1 

Do political incentives promote or inhibit corporate social responsibility? The role of local officials’ tenure 

Dear Dr. Wu:

I'm pleased to inform you that your manuscript has been deemed suitable for publication in PLOS ONE. Congratulations! Your manuscript is now with our production department. 

Kind regards, 

on behalf of

Dr. Vu Quang Trinh 

Academic Editor

PLOS ONE